# DIFFUSION GUIDANCE IS A CONTROLLABLE POLICY IMPROVEMENT OPERATOR

## ABSTRACT

At the core of reinforcement learning is the idea of learning beyond the performance in the data. However, scaling such systems has proven notoriously tricky. In contrast, techniques from generative modeling have shown to be remarkably scalable and are simple to train. In this work, we combine these strengths, by deriving a direct relation between policy improvement and guidance of diffusion models. The resulting framework, CFGRL, is a policy improvement operator that is trained with the simplicity of supervised learning, yet is more effective than typically-used weighted policy extraction strategies. On offline RL tasks, we observe a reliable trend—increased guidance weighting leads to increased performance. Additionally, the CFGRL framework can be adapted to "directly" extract policies from offline data *without* running a full end-to-end RL algorithm, allowing us to generalize simple supervised methods (e.g. goal-conditioned behavior cloning) to further prioritize optimality, gaining performance across the board without additional cost.

## 1 INTRODUCTION

Reinforcement learning (RL) provides a powerful framework for autonomous agents to attain strong performance by directly optimizing for task rewards. However, scaling up RL algorithms has proven notoriously challenging, particularly when using off-policy datasets. In contrast, modern generative modeling techniques have proven remarkably scalable, and have been used for related problems such as behavioral cloning (BC) (Janner et al., 2022; Black et al., 2024a; Chi et al., 2023). Can we leverage expressive generative modeling tools to derive simple, scalable RL techniques?

At the heart of RL is the idea of optimizing *beyond the performance shown in the data*. This is especially important when training agents from offline data that may have been collected by an exploratory or otherwise suboptimal policy. On one end of the spectrum, behavioral cloning methods are simple and can leverage stable generative modeling tools like diffusion (Ho et al., 2020; Lipman et al., 2024b) and flow-matching, but are only as optimal as the data. On the other end, iterative temporal difference techniques are in principle more optimal, but in practice can suffer from hyperparameter sensitivity and instability (Park et al., 2024; Kumar et al., 2020; Fujimoto et al., 2018) that have made it challenging to scale to larger tasks.

In this work, we combine the strengths of both settings by developing a framework which is trained with the simplicity of behavioral cloning, yet can further improve on the data behaviors. We first define policies as products of two factors – a prior reference policy, and an "optimality" distribution. When the optimality distribution is proportional to a monotonically increasing function of advantage, we prove that the resulting product will be an improvement over the prior.

The key insight is that we can sample from this product distribution via techniques from diffusion modeling, and we can do so in a straightforward and controllable way. Rather than optimizing an optimality *predictor*, we instead learn an equivalent optimality-*conditioned* policy, as done in classifier-free guidance (Ho & Salimans, 2022). The prior and conditional factors can then by dynamically combined during sampling, allowing for a degree of policy improvement that can be controlled *during test time*, without the need for retraining.

Our framework, which we refer to as CFGRL, provides a principled and powerful connection between generative modeling and policy improvement. Diffusion and flow-matching models already represent some of the most powerful approaches for imitation learning, but typically do not make use of guidance (Black et al., 2024a; Chi et al., 2023). CFGRL bridges a connection between guidance and

traditional RL objectives—in fact, under certain choices, guided sampling results in a distribution that is equivalent to the solution of a KL-constrained policy improvement objective.

We experimentally show applications of CFGRL in two settings. In the first, CFGRL is used as the policy extraction step of an end-to-end offline RL method, operating as a drop-in replacement for the standard weighted regression strategy. CFGRL provides a consistent improvement over the baseline. Scaling trends highlight how increasing the guidance term results in stronger policies *beyond* the point where weighted regression collapses, demonstrating how CFGRL can make better use of learned value functions.

In the second setting, we use the CFGRL framework to generalize goal-conditioned behavior cloning, noting that CFGRL can be used to controllably *extrapolate* the policies defined by the offline data, unlocking further performance gains without the need for additional training. CFGRL reliably outperforms baselines across the board on state-based, visual, and hierarchical settings, at times increasing success rates by a factor of two.

Our contributions are twofold. First, we propose a *principled connection* between diffusion model guidance and policy improvement in reinforcement learning. Second, we develop a set of simple *practical algorithms* that utilize the above connection, demonstrating reliably improvements both as a policy improvement operator in an end-to-end RL algorithm, and as a standalone replacement for naive goal-conditioned behavior cloning. We *do not claim to provide a full end-to-end RL algorithm*, but rather a powerful tool in the algorithm designer's toolbox.

## 2 RELATED WORK

**Offline RL.** Unlike standard RL, which involves exploring an environment, offline RL aims to learn a reward-maximizing policy solely from a previously collected dataset. The key challenge is to improve performance while preventing erroneous extrapolation when deviating too far from the dataset. Previous works target this problem from the value learning (Kumar et al., 2020; Kostrikov et al., 2022; Xu et al., 2023; Garg et al., 2023; An et al., 2021; Nikulin et al., 2023), and policy extraction (Wu et al., 2019; Fujimoto & Gu, 2021; Tarasov et al., 2023; Park et al., 2025b; Peng et al., 2019; Nair et al., 2020; Chen et al., 2023; Hansen-Estruch et al., 2023) directions. Our work most closely relates to weighted regression (Peng et al., 2019; Nair et al., 2020) and return-conditioned behavioral cloning (Kumar et al., 2019; Chen et al., 2021; Yamagata et al., 2023) methods (of which goal-conditioned hindsight relabeling (Andrychowicz et al., 2017; Ghosh et al., 2021; Emmons et al., 2022; Eysenbach et al., 2022a) is a special case), due to their emphasis on simple supervised objectives. However, we instead frame the tradeoff between regularization and policy improvement in terms of guiding a diffusion model, providing a way to *control* this tradeoff during test time.

**Diffusion and flow policies for RL.** Previous works have proposed diverse ways to leverage the expressivity of iterative generative models, such as diffusion (Sohl-Dickstein et al., 2015; Ho et al., 2020) and flow models (Lipman et al., 2023; Liu et al., 2023; Albergo & Vanden-Eijnden, 2023), to enhance the capabilities of RL policies. The main challenge with diffusion policy learning lies in *how to extract* (Park et al., 2025b): a diffusion policy to maximize the learned Q-function. Prior works propose strategies based on weighted regression (Lu et al., 2023; Kang et al., 2023; Ding et al., 2024; Zhang et al., 2025), reparameterized gradients (Wang et al., 2023; He et al., 2023; Ding & Jin, 2024; Ada et al., 2024; Zhang et al., 2024; Park et al., 2025b), rejection sampling (Chen et al., 2023; Hansen-Estruch et al., 2023; He et al., 2024), and more (Yang et al., 2023; Kuba et al., 2023; Mao et al., 2024; Chen et al., 2024b;a; Psenka et al., 2024; Chen et al., 2024c; Li et al., 2024; Mark et al., 2024; Fang et al., 2025; Ren et al., 2025). Our method introduces classifier-free guidance as a policy extraction mechanism. This has multiple benefits over previous approaches: unlike reparameterized gradient-based methods, it does not require (potentially unstable) backpropagation through time (Park et al., 2025b); unlike rejection sampling, it does not involve a costly sampling-then-filtering procedure. Close in spirit are Janner et al. (2022) and Ajay et al. (2022), which similarly utilize diffusion for decision making, however these methods are akin to world models as they operate over state trajectories and not actions. Algorithmically, the closest work to ours is Kuba et al. (2023), which also employs guidance over advantage-conditioned diffusion policies. Unlike this work, our framework supports a range of optimality functions rather than only $A = 0$, and does not rely on further rejection sampling. Additionally, we do *not* necessarily require an explicit value function to perform policy improvement—in Section 6, we further improve a goal-conditioned BC policy without additionally training value functions, whereas all aforementioned techniques would require doing so.

## 3 PRELIMINARIES

We assume a Markov decision process with state space $\mathcal{S}$, action space $\mathcal{A}$, a transition function $p(s' \mid s, a) : \mathcal{S} \times \mathcal{A} \to \Delta(\mathcal{S})$, reward function $r(s) : \mathcal{S} \to \mathbb{R}$, and initial state distribution $p(s_0) \in \Delta(\mathcal{S})$. We assume the state is fully observable. Our policy is a probability distribution over actions $\pi(a \mid s) : \mathcal{S} \to \Delta(A)$ that together with the environment defines a distribution of state-action trajectories $\tau = (s_0, a_0, s_1, a_1, \ldots)$. The standard RL objective is to learn a parameterized policy $\pi_\theta$ that maximizes the expected sum of future discounted rewards along such trajectories:

$$J(\pi_\theta) = \mathbb{E}_{\tau \sim p(\tau \mid \pi_\theta)} \sum_t \gamma^t r(s_t, a_t), \tag{1}$$

where $p(\tau \mid \pi_\theta)$ is defined as $p(s_0) \prod_{t=0}^\infty p(s_{t+1} \mid s_t, a_t) \pi_\theta(a_t \mid s_t)$.

A *policy improvement operator* is an update from a reference policy $\hat{\pi}$ to a new policy $\pi$ such that the RL objective above does not decrease: $J(\hat{\pi}) \leq J(\pi)$. This concept is formalized via $V_{\hat{\pi}}(s)$ and $Q_{\hat{\pi}}(s, a)$, which denote the discounted expected future reward under the reference policy starting from a given state or state-action pair, respectively (Sutton & Barto, 2005). The difference between these terms is the advantage, $A_{\hat{\pi}}(s, a) = Q_{\hat{\pi}}(s, a) - V_{\hat{\pi}}(s)$. A classic result shows that any update with non-negative advantage under the resulting state distribution, such that

$$\mathbb{E}_{(s,a) \sim p_\pi(s,a)}[A_{\hat{\pi}}(s, a)] \geq 0, \tag{2}$$

with $p_\pi(s, a)$ denoting the discounted state-action occupancy distribution, results in policy improvement (Schulman et al., 2015). In practice, we require algorithms that operate over samples from a previous reference policy. Therefore, practical algorithms will approximate the above condition under the previous policy's state distribution $p_{\hat{\pi}}(s)$, and aim to maximize:

$$\tilde{J}(\pi) = \mathbb{E}_{s \sim p_{\hat{\pi}}(s)}[\mathbb{E}_{a \sim \pi(a \mid s)}[A_{\hat{\pi}}(s, a)]]. \tag{3}$$

Prior work has shown that, as long as the divergence between the reference and resulting policy is bounded, the approximate objective in Equation (3) provides a bound on the true objective in Equation (2), enabling monotonic incremental improvement (Schulman et al., 2015).

To account for this divergence, it is common to utilize trust-region methods during policy improvement (Schulman et al., 2015; 2017). While various divergence measures have been considered (Sikchi et al., 2024), a standard choice is to penalize the KL divergence between the reference and resulting policy, resulting in the following KL-penalized RL objective as parameterized by a constant $\beta$:

$$J(\pi_\theta) = \mathbb{E}_{\tau \sim p(\tau \mid \pi_\theta)} \left[ \sum_t \gamma^t r(s_t, a_t) \right] - \beta \mathbb{E}_{s \sim p_\pi(s)} \left[ D_{\mathrm{KL}}(\pi_\theta(a \mid s) \parallel \hat{\pi}(a \mid s)) \right]. \tag{4}$$

One strategy to optimize the above objective is via iteratively applying a policy gradient (Sutton et al., 1999); however, such methods require on-policy samples and can have high variance. The community has instead often relied on methods that resemble supervised learning, such as weighted regression (Peters & Schaal, 2007; Peng et al., 2019). In the following sections, we introduce a policy improvement strategy based on generative modeling that maintains the simplicity of supervised learning, yet allows for controllable policy improvement.

## 4 DIFFUSION GUIDANCE IS A CONTROLLABLE POLICY IMPROVEMENT OPERATOR

In this work, we establish a connection between classifier-free guidance in diffusion modeling and policy improvement in RL. We use this relation to develop a simple framework, CFGRL, which enables us to leverage stable diffusion training methods, while allowing the degree of policy improvement to be controlled at *test time*, rather than having to be decided during training.

**Product policies.** We begin by parameterizing policies as a product of two factors—a reference policy, and an *optimality* function $f : \mathbb{R} \to \mathbb{R}$, which is conditional on advantage:

$$\pi(a \mid s) \propto \hat{\pi}(a \mid s) \, f(A(s, a)). \tag{5}$$

The motivation behind this factorization is to frame improved policies in terms of a probabilistic adjustment to the current reference policy, building on the control-as-inference framework (Levine, 2018). As we will show later, diffusion guidance is naturally suited to sampling from such distributions.

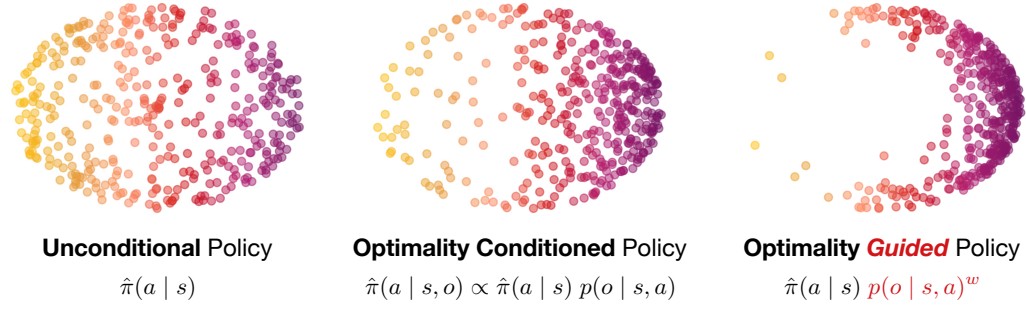

**Unconditional** Policy  $\qquad$ **Optimality Conditioned** Policy $\qquad$ **Optimality *Guided*** Policy

$$\hat{\pi}(a \mid s) \qquad \hat{\pi}(a \mid s, o) \propto \hat{\pi}(a \mid s)\, p(o \mid s, a) \qquad \hat{\pi}(a \mid s)\, p(o \mid s, a)^{w}$$

Figure 1: **While conditioning on optimality can create a baseline level of improvement, policies can be *further* improved by attenuating this conditioning.** When $p(o \mid s, a)$ is proportional to a monotonically increasing function of advantage, then attenuation provably increases expected return, and this can be accomplished naturally with diffusion guidnace.

We now show how product policies can be used as *policy improvement operators*. When $f$ fulfills certain common criteria, the resulting policy will provably be an improvement over the reference:

**Remark 1** (**Improvement of product policies**). *If $f$ is a non-negative, monotonically increasing function of $A_{\hat{\pi}}(s, a)$, then the product $\pi(a \mid s) \propto \hat{\pi}(a \mid s) f(A(s, a))$ is an improvement over $\hat{\pi}(a \mid s)$.*

We provide a formalization and proof of this claim in the Appendix (Theorem 1), which generalizes previous results with bandits (Dayan & Hinton, 1997; Peters & Schaal, 2007) and exponentiated advantages (Peng et al., 2019).

The above finding reveals a simple path towards policy improvement. If we can sample from a properly weighted product policy, then the resulting policy will achieve a higher expected return then the reference.

Crucially, we can control the *degree* of improvement by sampling from an attenuated optimality function. To do so, we consider product policies where the optimality functions are exponentiated:

**Remark 2** (**Further improvement via attenuation**). *Let $0 \leq w_1 < w_2$ be real numbers, and $\pi_{w_i} \propto \hat{\pi}(a \mid s) \, f(A(s, a))^{w_i}$ for $i = 1, 2$. Then, $\pi_{w_2}$ is an improvement over $\pi_{w_1}$.*

The proof of the above theorem is again provided in the Appendix (Theorem 2). Of course, there is no free lunch. While a higher exponent leads to an improved policy in terms of $A_{\hat{\pi}}$, the resulting policy is also further deviated from the reference. Thus, the empirical performance of an over-adjusted policy may suffer due to a distribution shift.

This tradeoff between adhering to the reference policy and maximizing return can be understood via the KL-regularized RL objective in Equation (4). Notably, the *solutions* to the mentioned objective naturally form a set of product policies:

**Remark 3** (**KL-regularized reward-maximization results in product policies** (Peng et al., 2019)). *The policies that maximize Equation (4) under a given KL-penalty $\beta$ take the form of*

$$\pi(a \mid s) \propto \hat{\pi}(a \mid s) \exp(A(s, a))^{1/\beta}. \tag{6}$$

Equivalently, the $\beta$ term can be folded into the exponent, e.g. $\pi(a \mid s) \propto \hat{\pi}(a \mid s) \exp(A(s, a)/\beta)$. This condensed objective has been used in prior works (Peters & Schaal, 2007; Peng et al., 2019) to directly learn $\pi$; however, in their methods the $\beta$ hyperparameter must be specified ahead of time. As shown in the next sections, we will instead develop a framework where the product factors are represented *independently*, allowing their composition to be freely controlled during evaluation time.

### 4.1 COMPOSING FACTORS VIA DIFFUSION GUIDANCE

Having understood that product policies are a natural way to induce policy improvement, we can now instantiate such policies using machinery from diffusion modeling. We start by casting the optimality function as a binary random variable[1] $o \in \{\varnothing, 0, 1\}$ whose likelihood is defined via $f$:

$$p(o \mid s, a) = f(A(s, a))/Z(s) \tag{7}$$

---

[1]Slightly abusing notation, we abbreviate $o = 1$ as $o$ when it is clear from context. $o = \varnothing$ represents an unconditional case where $o$ is not specified, i.e., the union of $o = 0$ and $o = 1$.

| **Algorithm 1** CFGRL Training | **Algorithm 2** CFGRL Sampling |
|---|---|
| **while** not converged **do** | $a \sim \mathcal{N}(0, I)$ |
|    Collect data, or use offline data. | $t \leftarrow 0$ |
|    $(s, a) \sim D, a_0 \sim \mathcal{N}(0, I), t \sim U(0, 1)$ | **for** $n \in [0, \dots, N-1]$ **do** |
|    Label with optimality $o \in \{0, 1\}$. |    $v = (1 - w) \, v_\theta(a, t, s, \varnothing) + w \, v_\theta(a, t, s, o = 1)$ |
|    If rand() $< 0.1$, set optimality $o = \varnothing$. |    $a \leftarrow a + (n/N)v$ |
|    $a_t \leftarrow (1 - t) \, a_0 + t \, a$ |    $t \leftarrow t + (n/N)$ |
|    $\theta \leftarrow \nabla_\theta \| v_\theta(a_t, t, s, o) - (a - a_0) \|^2$ | **end for** |
| **end while** | **return** $a$ |

where $Z(s) = \int f(A(s, a'))da'$ is a state-dependent normalization factor. We will not require estimating $Z(s)$. The product policy from Equation (5) can now be equivalently defined as:

$$\pi(a \mid s) \propto \hat{\pi}(a \mid s) \, p(o \mid s, a). \tag{8}$$

Recall that diffusion models implicitly model a distribution by learning its normalized *score function*, i.e., the gradient of log-likelihood under that distribution (Song & Ermon, 2019). Score functions have the useful property that for product distributions, they are additively composable. As such, the score of the product policy above can be represented as the sum of two factors:

$$\nabla_a \log \pi(a \mid s) = \nabla_a \log \hat{\pi}(a \mid s) + \nabla_a \, \log p(o \mid s, a). \tag{9}$$

**Avoiding an explicit optimality predictor.** In many cases, we would rather avoid explicitly learning the $p(o \mid s, a)$ distribution. For one thing, optimality must hold a valid probability distribution, and calculating the normalization term $Z(s)$ can be tricky. Secondly, explicitly backpropagating through a neural network predictor may result in adversarial gradient attacks (Goodfellow et al., 2015) especially at out-of-distribution actions (Kostrikov et al., 2022; Kumar et al., 2020). We also note that if one wanted to learn an optimality predictor, it would need to remain accurate under the support of *partially-noised* actions.

Instead, we can utilize an insight from classifier-free guidance (Ho & Salimans, 2022), and use Bayes' rule to invert the optimality distribution into an optimality-*conditioned* policy score function:

$$\nabla_a \log \pi(a \mid s) = \nabla_a \log \hat{\pi}(a \mid s) + (\nabla_a \log \hat{\pi}(a \mid s, o) - \nabla_a \log \hat{\pi}(a \mid s)). \tag{10}$$

With the above form, both factors can be unified into a single conditional policy. We can represent both factors with the same neural network, and train via a straightforward diffusion modeling objective.

**Guidance controls the attenuation of optimality.** A key benefit of defining the product distribution in terms of composable factors is that the ratio between these factors can be dynamically controlled. Introducing a guidance weight $w$, the score function

$$\nabla_a \log \hat{\pi}(a \mid s) + w \, (\nabla_a \log \hat{\pi}(a \mid s, o) - \nabla_a \log \hat{\pi}(a \mid s)) \tag{11}$$

corresponds to the attenuated distribution

$$\pi(a \mid s) \propto \hat{\pi}(a \mid s) \, p(o \mid s, a)^w, \text{ and equivalently } \hat{\pi}(a \mid s) f(A(s, a))^w. \tag{12}$$

Recall that in Remark 2, we showed that increasing $w$ results in further policy improvement. This implies a simple yet crucial relationship—by controlling the guidance weight during sampling, we can sample from product policies that *controllably* improve on the reference policy (at the cost of adherence to the prior). The tradeoff is a key hyperparameter to tune in offline RL (Park et al., 2024), and often requires many sweeps. In contrast, with CFGRL, this sweep can be performed at test-time over a *single* network, without the need for retraining.

### 4.2 TRAINING AND SAMPLING WITH CFGRL

We instantiate a single diffusion network to serve as both the conditional and unconditional policy. For simplicity, we adopt the flow-matching framework for training. While flow networks predict velocity rather than score, previous works have shown they retain similar properties in practice (Gao et al., 2024; Lipman et al., 2024a). The policy is modeled by a velocity field $v_\theta$ conditioned on a partially-noised action $a_t$, along with a noise scale $t$, the current state $s$, as well as the previously defined *optimality variable* $o \in \{\varnothing, 0, 1\}$. This network is trained via the following loss function:

$$\mathcal{L}(\theta) = \mathbb{E}_{s, a \sim D} \left[ \| v_\theta(a_t, t, s, o) - (a - a_0) \|^2 \right] \quad \text{where} \quad a_t = (1 - t)a_0 + ta, \tag{13}$$

and where the noise scale $t$ is sampled uniformly between $[0, 1]$, and $a_0 \sim N(0, 1)$ is Gaussian noise.

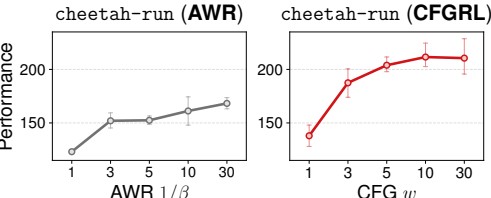

Figure 3: **CFGRL controls the tradeoff between reference adherence and optimality by adjusting the guidance weighting, and does so better than AWR.** Guidance addresses the same motivation behind tuning the temperature in advantage-weighted regression, however, it can be tuned during *test time* rather than via retraining, and empirically leads to a higher maximum performance.

Table 1: **ExORL results.**

| Task | AWR | CFGRL |
|---|---|---|
| walker-stand | $603_{\pm 8}$ | $\mathbf{782}_{\pm 8}$ |
| walker-walk | $444_{\pm 4}$ | $\mathbf{608}_{\pm 32}$ |
| walker-run | $247_{\pm 10}$ | $\mathbf{282}_{\pm 6}$ |
| quadruped-walk | $\mathbf{776}_{\pm 15}$ | $762_{\pm 25}$ |
| quadruped-run | $485_{\pm 7}$ | $\mathbf{571}_{\pm 25}$ |
| cheetah-run | $168_{\pm 7}$ | $\mathbf{216}_{\pm 15}$ |
| cheetah-run-backward | $146_{\pm 8}$ | $\mathbf{262}_{\pm 26}$ |
| jaco-reach-top-right | $33_{\pm 2}$ | $\mathbf{72}_{\pm 6}$ |
| jaco-reach-top-left | $30_{\pm 8}$ | $\mathbf{46}_{\pm 6}$ |

Table 2: **OGBench results.**

| Task | AWR | CFGRL |
|---|---|---|
| pointmaze-large-navigate | $70_{\pm 25}$ | $\mathbf{100}_{\pm 0}$ |
| pointmaze-teleport-navigate | $3_{\pm 7}$ | $\mathbf{57}_{\pm 7}$ |
| antmaze-large-navigate | $\mathbf{50}_{\pm 9}$ | $20_{\pm 9}$ |
| antmaze-teleport-navigate | $22_{\pm 19}$ | $\mathbf{30}_{\pm 22}$ |
| humanoidmaze-large-navigate | $\mathbf{3}_{\pm 4}$ | $0_{\pm 0}$ |
| antsoccer-arena-navigate | $7_{\pm 0}$ | $\mathbf{20}_{\pm 5}$ |
| cube-single-play | $\mathbf{85}_{\pm 8}$ | $82_{\pm 3}$ |
| scene-play | $\mathbf{18}_{\pm 3}$ | $17_{\pm 9}$ |
| puzzle-3x3-play | $\mathbf{3}_{\pm 7}$ | $\mathbf{3}_{\pm 4}$ |

## 5 CFGRL IMPROVES OVER WEIGHTED POLICY EXTRACTION IN OFFLINE RL

A common approach to offline RL is to learn a state-action value function $Q_\theta(s, a)$, and then extract a policy from this value function via a regularized *policy extraction* method that stays close to the behavior policy while maximize the value function. This regularization is critical to avoid out-of-distribution actions for which the value function is likely to overestimate the value (Peng et al., 2019; Kostrikov et al., 2022). *Weighted regression* methods are a particularly simple class of methods for doing this, with advantage-weighted regression (AWR) or its variants being a common choice in recent work (Peng et al., 2019; Nair et al., 2020; Wang et al., 2020). AWR is trained in a fully supervised manner, and does not require querying the values of non-dataset actions, with the training objective given by

$$J_{\text{AWR}}(\theta) = E_{(s,a) \sim D} \left[ \log \pi_\theta(a \mid s) \exp(A(s, a) \times (1/\beta)) \right], \tag{14}$$

where $A(s, a) = Q(s, a) - V(s)$ is calculated as the difference of learned $Q_\theta(s, a)$ and $V_\theta(s)$ networks, and $\beta$ is a temperature hyperparameter.

However, a weakness of AWR is that the weightings can become peaked, such that much of the data in training is not used effectively, resulting in a weak learning signal. This phenomenon is shown in Figure 2 (left), which plots the magnitudes of per-element gradients within an AWR batch. Notably, the magnitudes are dominated by a few outlier state-action pairs that hold particularly high weights. In this case, the rest of the batch is effectively ignored, and AWR derives the gradient only from a small subset of the available data.

We now show how we can instead utilize CFGRL to alleviate this issue. Specifically, we will instantiate CFGRL with a particularly simple optimality criteria:

$$o = \begin{cases} 1 & \text{if } A(s, a) \geq 0 \\ 0 & \text{if } A(s, a) < 0, \end{cases} \tag{15}$$

which nicely lends itself to Bayes' inversion for Equation (10). Equivalently, we are assigning $f = \mathbf{1}(A \geq 0)$ which is both non-negative and non-decreasing, fulfilling the criteria of Remark 1.

The end-to-end procedure with CFGRL is simple. Given an state-action sample $(s, a) \sim D$, we label

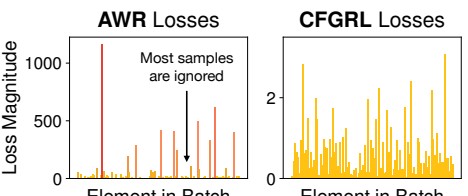

Figure 2: **Weighted regression methods result in uneven gradient magnitudes within a batch.** This can limit the effective signal that each batch provides. In contrast, CFGRL uses a simple conditional diffusion modeling loss with even weighting.

that pair with $o \in \{0, 1\}$ according the above criteria, utilizing our current value function. We then train with the standard conditional diffusion-modeling loss as done in Section 4.2. Notably, there is *no weighting term used in training*. As such, gradients within each batch remain reasonably distributed, as shown in Figure 2 (right).

There is a suggestive similarity between the temperature hyperparameter in AWR and the guidance weight in CFGRL. In fact, both these parameters play the same role—they control the tradeoff between adherence to a reference policy and maximization of rewards. However, with AWR, the temperature must be chosen beforehand and is folded into the optimization objective. CFGRL keeps the prior policy and the optimality-conditioned policy separate, and only combines them during sampling. Thus, this tradeoff can be adjusted *without retraining* when using CFGRL, making it easy to find the best value.

Furthermore, we will see that the guidance term in CFGRL is empirically more effective than the temperature of AWR. Note that while the absolute scale of $w$ and $(1/\beta)$ can vary, they have a proportional relationship and share the same base case at $w = (1/\beta) = 0$, in which case the resulting policy simply mimics the dataset policy. We examine the scaling of performance over different guidance and temperature values in Figure 3. For AWR, performance saturates around a temperature of $(1/\beta) = 10$. In contrast, guidance continues to improve beyond performance beyond this point, displaying a longer-lasting trend.

**Experimental comparison.** We further establish the comparison between AWR and CFGRL on 9 tasks from the ExORL benchmark (Yarats et al., 2022), which contains data collected by an exploratory agent, along with 9 single-task environments from the OGBench suite (Park et al., 2025a). In all experiments, we use the *same* state-action value function trained via implicit Q-learning for both methods (Kostrikov et al., 2022), which notably does not require a policy in the loop to learn and is therefore independent of the extraction method that is used downstream. For AWR, we sweep over $1/\beta$ in the set of $\{1, 3, 10, 30\}$, and for CFGRL, we sweep over $w \in \{1, 1.25, 1.5, 2.0, 3.0\}$. Results are presented in Tables 1 and 2, and are averaged over 4 seeds. On a strong majority of tasks, CFGRL achieves a better final performance than AWR. This indicates that policy extraction with CFGRL, which also corresponds to a simple generative modeling objective, is more effective than the widely used AWR method.

## 6 CFGRL UNLOCKS HIDDEN GAINS IN GOAL-CONDITIONED BC

While the overall CFCRL framework is general, it is particularly appealing in the special-case of goal-conditioned RL, where it is common to use a simple (though crude) approximation that bypasses the need for a learned value estimator (Eysenbach et al., 2022a; Ghosh et al., 2021; Black et al., 2024b). In such settings, the objective is to find a goal-conditioned policy $\pi(a \mid s, g) : \mathcal{S} \times \mathcal{S} \rightarrow \Delta(\mathcal{A})$ that maximizes likelihood of reaching the goal:

$$J(\pi) = \mathbb{E}_{\tau \sim p(\tau | \pi), \ g \sim p(g)} \left[ \sum_t \gamma^t \delta_g(s_t) \right], \tag{16}$$

where $p(g) \in \Delta(\mathcal{S})$ is a goal distribution and reward is $\delta_g$, i.e. the Dirac delta "function" at $g$.[2] While in principle, we can optimize the above objective with a full RL procedure, an often-used simplification is to perform goal-conditioned *behavioral cloning* (GCBC), which maximizes:

$$J_{\text{GCBC}}(\theta) = \mathbb{E}_{(s_t, a_t) \sim \mathcal{D}, \ \Delta \sim \text{Geom}(1-\gamma)} [\log \pi_\theta(a_t \mid s_t, s_{t+\Delta})], \tag{17}$$

where $\text{Geom}(1 - \gamma)$ denotes the Geometric distribution with parameter $1 - \gamma$, and we often denote $s_{t+\Delta}$ as $g$. This avoids the need to train a value function. GCBC can be seen as a special case of conditional behavioral cloning methods that filter for actions that empirically reach a future goal. While the GCBC objective above is simple, it does not converge to the optimal goal-reaching policy, especially when the dataset is suboptimal (Eysenbach et al., 2022a; Ghugare et al., 2024)

**Generalizing past naïve GCBC.** Based on our CFGRL framework in Section 4, we now introduce a method to further improve GCBC policies *while still avoiding training value functions*. The key insight is that, since CFGRL enables one step of policy improvement over the base policy, applying

---

[2]This "function" is well-defined in a discrete state space, but requires a measure-theoretic formulation (Touati & Ollivier, 2021) to be well-defined in a continuous state space, which we omit for simplicity.

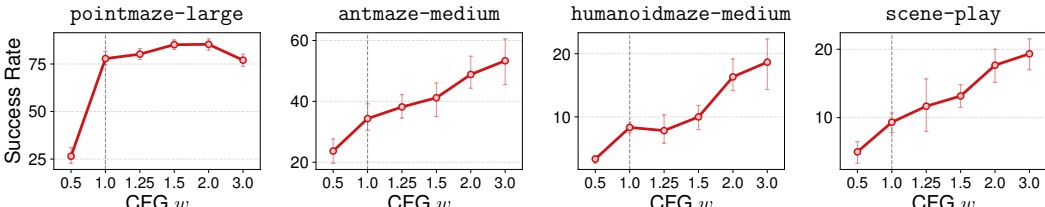

Figure 4: **CFGRL can extrapolate *beyond* the GCBC policy, unlocking further performance gains.** In fact, GCBC is implicitly a special case of the CFGRL policy where $w = 1$. By instead considering $w > 1$, the resulting policy is an improvement over the original. We show that performance steadily increases with $w$ on a range of environments.

CFGRL with guidance on the goal $g$ will produce a policy that is better than the standard GCBC policy. While this policy is still not as optimal as the end-to-end GCRL solution (which in general would require many steps of policy improvement), prior work has shown that even one step of policy improvement can lead to significant performance gains in a range of RL settings (Brandfonbrener et al., 2021; Eysenbach et al., 2022b). Indeed, our experiments will confirm that CFGRL enables a significant increase in performance over standard GCBC.

We begin by noting that the GCBC policy that maximizes Equation (17) is given as follows (Eysenbach et al., 2022b):

$$\pi(a \mid s, g) = \frac{\hat{\pi}(a \mid s) p^\gamma(g \mid s, a)}{p^\gamma(g \mid s)} \propto \hat{\pi}(a \mid s) \, Q_{\hat{\pi}}(s, a, g), \tag{18}$$

where $p^\gamma$ denotes the distribution induced by the Geometric goal sampling procedure.

Next, we note that the second factor in Equation (18) satisfies the conditions of Remark 1 (i.e., a bounded, non-negative, non-decreasing function in $A_{\hat{\pi}}(s, a)$).[3] Thus, we can invoke Remark 2 to achieve a policy improvement. Specifically, the resulting policy $\pi(a \mid s) \propto \hat{\pi}(a \mid s) \, p(g \mid s, a)^w$ under any exponent $w \geq 1$ will result in a policy improvement, and we can sample from the attenuated second factor via guidance:

$$\nabla_a \log \hat{\pi}(a \mid s) + w \, (\nabla_a \log \pi(a \mid s, g) - \nabla_a \log \hat{\pi}(a \mid s)). \tag{19}$$

The above CFGRL interpretation reveals a simple recipe for improving beyond the GCBC policy. Specifically, naive GCBC results in a product policy that implicitly assumes a weighting of $w = 1$. Guidance allows us to instead consider $w > 1$, leading to improved performance.

We emphasize the practical benefits of this setup—improvement can be gained for "free" relative to a standard GCBC setup. The components of Equation (19) are simply the original goal-conditioned BC policy $\pi(a \mid s, g)$ along with an unconditional BC policy $\hat{\pi}(a \mid s)$. We do *not* require any additional techniques such as training an explicit value function, or sampling further on-policy actions.

## 6.1 EXPERIMENTAL RESULTS

**Tasks.** We empirically verify effectiveness over 17 state-based and 7 pixel-based goal-conditioned RL tasks from the OGBench task suite (Park et al., 2025a) (Figure 5). These tasks span a variety of robotic navigation and manipulation domains, including whole-body humanoid control, maze navigation, sequential object manipulation, and combinatorial puzzle solving. Among them, the tasks prefixed with "visual-" require image-based control.

**Methods.** In this experiment, we consider four imitation learning baselines that do not involve value learning (recall that CFGRL also does not train a value function): (1) BC, (2) flow BC (Chi et al., 2023), (3) goal-conditioned BC (GCBC) (Ghosh et al., 2021), and (4) flow GCBC. BC trains an (unconditional) behavioral cloning policy, and GCBC trains a goal-conditioned policy with Equation (17). Flow BC and flow GCBC maximize the same objectives, but with flow policies (Chi et al., 2023; Black et al., 2024a). In the CFGRL framework, flow BC corresponds to CFGRL with $w = 0$ and flow GCBC corresponds to $w = 1$.

---

[3]Specifically, we set $f$ as $Q_{\hat{\pi}}(s, a, g)$, which is a bounded, non-negative, non-decreasing function of $A_{\hat{\pi}}(s, a, g)$. This can also be understood as defining $p(o \mid g, s, a) \propto p^\gamma(g \mid s, a)$, i.e., an action's optimality is proportional to the likelihood of reaching the goal in the discounted future.

Table 3: **Improving on GCBC.** CFGRL consistently improves performance over GCBC across the board. Numbers at or above the $95\%$ of the best performance in each category are boldfaced, as in OGBench.

| Task | Flat Policies | | | | | Hierarchical Policies | | |
|---|---|---|---|---|---|---|---|---|
| | BC | Flow BC | GCBC | Flow GCBC | CFGRL | HGCBC | Flow HGCBC | HCFGRL |
| pointmaze-medium-navigate | $0_{\pm0}$ | $1_{\pm2}$ | $9_{\pm5}$ | $66_{\pm4}$ | $\mathbf{77}_{\pm6}$ | $0_{\pm0}$ | $57_{\pm7}$ | $\mathbf{63}_{\pm5}$ |
| pointmaze-large-navigate | $0_{\pm0}$ | $0_{\pm0}$ | $25_{\pm9}$ | $\mathbf{74}_{\pm7}$ | $\mathbf{77}_{\pm5}$ | $0_{\pm0}$ | $\mathbf{75}_{\pm6}$ | $57_{\pm8}$ |
| pointmaze-giant-navigate | $0_{\pm0}$ | $0_{\pm0}$ | $2_{\pm2}$ | $4_{\pm2}$ | $\mathbf{30}_{\pm10}$ | $0_{\pm0}$ | $6_{\pm5}$ | $\mathbf{18}_{\pm8}$ |
| pointmaze-teleport-navigate | $0_{\pm0}$ | $0_{\pm1}$ | $24_{\pm7}$ | $\mathbf{46}_{\pm4}$ | $41_{\pm8}$ | $6_{\pm5}$ | $\mathbf{37}_{\pm10}$ | $\mathbf{38}_{\pm13}$ |
| antmaze-medium-navigate | $13_{\pm2}$ | $15_{\pm6}$ | $25_{\pm8}$ | $42_{\pm9}$ | $\mathbf{53}_{\pm12}$ | $45_{\pm7}$ | $67_{\pm8}$ | $\mathbf{90}_{\pm5}$ |
| antmaze-large-navigate | $11_{\pm4}$ | $5_{\pm2}$ | $20_{\pm4}$ | $22_{\pm5}$ | $\mathbf{24}_{\pm10}$ | $45_{\pm4}$ | $61_{\pm5}$ | $\mathbf{78}_{\pm4}$ |
| antmaze-giant-navigate | $0_{\pm0}$ | $0_{\pm0}$ | $0_{\pm0}$ | $0_{\pm0}$ | $\mathbf{1}_{\pm1}$ | $8_{\pm5}$ | $14_{\pm5}$ | $\mathbf{38}_{\pm7}$ |
| antmaze-teleport-navigate | $4_{\pm2}$ | $3_{\pm2}$ | $19_{\pm5}$ | $24_{\pm7}$ | $\mathbf{35}_{\pm9}$ | $35_{\pm6}$ | $38_{\pm5}$ | $\mathbf{50}_{\pm5}$ |
| humanoidmaze-medium-navigate | $3_{\pm3}$ | $2_{\pm2}$ | $6_{\pm3}$ | $8_{\pm3}$ | $\mathbf{19}_{\pm6}$ | $13_{\pm3}$ | $23_{\pm4}$ | $\mathbf{64}_{\pm10}$ |
| humanoidmaze-large-navigate | $0_{\pm1}$ | $0_{\pm0}$ | $2_{\pm1}$ | $1_{\pm2}$ | $\mathbf{3}_{\pm2}$ | $8_{\pm5}$ | $11_{\pm2}$ | $\mathbf{38}_{\pm4}$ |
| antsoccer-arena-navigate | $2_{\pm2}$ | $4_{\pm1}$ | $4_{\pm3}$ | $10_{\pm5}$ | $\mathbf{15}_{\pm6}$ | $8_{\pm4}$ | $16_{\pm5}$ | $\mathbf{37}_{\pm5}$ |
| antsoccer-medium-navigate | $0_{\pm0}$ | $0_{\pm0}$ | $\mathbf{6}_{\pm4}$ | $5_{\pm3}$ | $5_{\pm3}$ | $7_{\pm2}$ | $8_{\pm3}$ | $\mathbf{11}_{\pm4}$ |
| cube-single-play | $4_{\pm2}$ | $7_{\pm5}$ | $6_{\pm2}$ | $8_{\pm2}$ | $\mathbf{11}_{\pm5}$ | $12_{\pm3}$ | $26_{\pm6}$ | $\mathbf{46}_{\pm6}$ |
| cube-double-play | $1_{\pm1}$ | $2_{\pm1}$ | $2_{\pm1}$ | $\mathbf{3}_{\pm1}$ | $2_{\pm2}$ | $2_{\pm1}$ | $21_{\pm7}$ | $\mathbf{42}_{\pm5}$ |
| scene-play | $5_{\pm2}$ | $5_{\pm3}$ | $4_{\pm3}$ | $15_{\pm5}$ | $\mathbf{19}_{\pm4}$ | $7_{\pm2}$ | $14_{\pm4}$ | $\mathbf{18}_{\pm5}$ |
| puzzle-3x3-play | $\mathbf{3}_{\pm2}$ | $2_{\pm2}$ | $\mathbf{3}_{\pm2}$ | $1_{\pm1}$ | $2_{\pm2}$ | $\mathbf{3}_{\pm2}$ | $1_{\pm1}$ | $2_{\pm2}$ |
| puzzle-4x4-play | $\mathbf{0}_{\pm0}$ | $\mathbf{0}_{\pm1}$ | $\mathbf{0}_{\pm0}$ | $\mathbf{0}_{\pm0}$ | $\mathbf{0}_{\pm0}$ | $\mathbf{0}_{\pm0}$ | $\mathbf{0}_{\pm0}$ | $\mathbf{0}_{\pm0}$ |
| visual-antmaze-medium-navigate | $14_{\pm5}$ | $7_{\pm1}$ | $11_{\pm5}$ | $19_{\pm4}$ | $\mathbf{23}_{\pm4}$ | - | - | - |
| visual-antmaze-large-navigate | $6_{\pm4}$ | $1_{\pm1}$ | $5_{\pm2}$ | $3_{\pm1}$ | $\mathbf{11}_{\pm2}$ | - | - | - |
| visual-cube-single-play | $1_{\pm1}$ | $1_{\pm1}$ | $1_{\pm2}$ | $13_{\pm7}$ | $\mathbf{37}_{\pm9}$ | - | - | - |
| visual-cube-double-play | $0_{\pm0}$ | $0_{\pm0}$ | $0_{\pm0}$ | $\mathbf{10}_{\pm4}$ | $7_{\pm5}$ | - | - | - |
| visual-scene-play | $5_{\pm1}$ | $7_{\pm2}$ | $5_{\pm2}$ | $25_{\pm4}$ | $\mathbf{40}_{\pm8}$ | - | - | - |
| visual-puzzle-3x3-play | $\mathbf{1}_{\pm2}$ | $0_{\pm1}$ | $0_{\pm1}$ | $0_{\pm1}$ | $0_{\pm1}$ | - | - | - |
| visual-puzzle-4x4-play | $\mathbf{0}_{\pm0}$ | $\mathbf{0}_{\pm0}$ | $\mathbf{0}_{\pm0}$ | $\mathbf{0}_{\pm0}$ | $\mathbf{0}_{\pm0}$ | - | - | - |

In addition to the five "flat" methods above, we also consider *hierarchical* behavioral cloning (Gupta et al., 2019; Lynch et al., 2019) on state-based tasks, where we train both a high-level policy $\pi^h(\ell \mid s, g) : \mathcal{S} \times \mathcal{S} \to \Delta(\mathcal{S})$ that outputs subgoals $\ell$ and a low-level policy $\pi^\ell(a \mid s, \ell) : \mathcal{S} \times \mathcal{S} \to \Delta(\mathcal{A})$ that takes subgoals and outputs actions. In this setting, we can apply CFGRL to each level's GCBC objective to enhance the optimality of both policies. We call this variant hierarchical CFGRL (HCFGRL). As baselines, we consider hierarchical GCBC Gupta et al. (2019); Lynch et al. (2019) and flow hierarchical GCBC, which trains Gaussian and flow policies, respectively.

**Results.** Table 3 presents evaluation results over 8 seeds for state-based tasks and 4 seeds for pixel-based tasks, denoting standard deviations after the $\pm$ symbols. Results show CFGRL consistently outperforms all four baselines on most of the tasks, even with a single fixed value of the guidance strength ($w = 3$). Notably, on some tasks (e.g., `pointmaze-giant` and `visual-cube-single`), CFGRL achieves more than $3\times$ the success of the strongest baseline. We emphasize that this improvement is achieved simply by contrasting the prior and GCBC policies, *without* training a value function. As in Section 5, we measure how performance varies with different values of guidance weights $w$. We present the results on four tasks in Figure 4 (see Figure 6 for the full results), which shows that the performance generally improves as $w$ increases, as predicted by Remark 2.

## 7 DISCUSSION AND CONCLUSION

In this work, we introduced a principled connection between diffusion guidance and policy improvement in RL. Using this connection, we derive a framework that combines the simplicity of generative modeling objectives with the policy improvement capabilities of RL. We then instantiate this framework as 1) a policy extraction method in offline RL when learning a value function, and 2) a "direct" method of learning performant policies *without* a value function. We show that CFGRL improves over the widely used AWR approach in the offline RL setting, and achieves a substantial improvement over GCBC in the goal-conditioned setting, while maintaining the simplicity of these prior methods.

**Limitations.** Our method does not claim to replace full RL procedures—we assume a given value function and do not make any prescriptions about how to train it. In our experiments, CFGRL takes the place of prior supervised learning methods for policy extraction, maintaining their simplicity and stability. However, more advanced policy extraction methods and online RL techniques, such as policy gradients (Lillicrap et al., 2016; Schulman et al., 2017), could provide for stronger extrapolation. By itself, CFGRL does not represent a state-of-the-art RL algorithm, but rather an additional tool in the algorithm designer's toolbox that can take the place of policy extraction methods such as AWR, as well as a theoretical connection that we hope will inspire future work.

REPRODUCIBILITY STATEMENT

We provide our anonymized implementation and instructions at https://anonymous.4open.science/r/cfgrl_submit-206C, and describe the full experimental details in Appendix.

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

## A  THEORETICAL RESULTS

**Lemma 1** (Chebyshev's sum inequality for probability measures)**.** *For any probability measure $\mu$ on $\mathbb{R}$ and any bounded, measurable, non-decreasing functions $g, h : \mathbb{R} \to \mathbb{R}$,*

$$\int_{\mathbb{R}} g(x)h(x)\mu(\mathrm{d}x) \geq \int_{\mathbb{R}} g(x)\mu(\mathrm{d}x) \int_{\mathbb{R}} h(x)\mu(\mathrm{d}x). \tag{20}$$

*Proof.* Since $g$ and $h$ are non-decreasing, the signs of $g(y) - g(z)$ and $h(y) - h(z)$ are the same for any $y, z \in \mathbb{R}$. Hence, we have

$$0 \leq \int_{\mathbb{R} \times \mathbb{R}} (g(y) - g(z))(h(y) - h(z))(\mu \otimes \mu)(\mathrm{d}y, \mathrm{d}z) \tag{21}$$

$$= \int_{\mathbb{R}} \left( \int_{\mathbb{R}} (g(y)h(y) + g(z)h(z) - g(y)h(z) - g(z)h(y))\mu(\mathrm{d}y) \right) \mu(\mathrm{d}z) \tag{22}$$

$$= 2 \int_{\mathbb{R}} g(x)h(x)\mu(\mathrm{d}x) - 2 \int_{\mathbb{R}} g(x)\mu(\mathrm{d}x) \int_{\mathbb{R}} h(x)\mu(\mathrm{d}x), \tag{23}$$

from which the conclusion follows, where $\mu \otimes \mu$ denotes the product measure of $\mu$ and itself, and we use Fubini's theorem in the second line. $\qquad\square$

**Lemma 2.** *Let $s \in \mathcal{S}$ be a state, $\pi, \hat{\pi} : \mathcal{S} \to \Delta(\mathcal{A})$ be policies, and $f : \mathbb{R} \to \mathbb{R}$ be a bounded, measurable, non-negative, non-decreasing function. Suppose that $\pi(a \mid s) = f(A_{\hat{\pi}}(s, a))\hat{\pi}(a \mid s)$ and $\mathbb{E}_{a \sim \hat{\pi}(\cdot|s)}[f(A_{\hat{\pi}}(s, a))] = 1$. Then,*

$$\mathbb{E}_{a \sim \pi(\cdot|s)}[Q_{\hat{\pi}}(s, a)] \geq V_{\hat{\pi}}(s). \tag{24}$$

*Proof.* To apply Lemma 1, we first rewrite the left-hand side of Equation (24) using probability measures as follows:

$$\mathbb{E}_{a \sim \pi(\cdot|s)}[Q_{\hat{\pi}}(s, a)] = \int_{\mathcal{A}} Q_{\hat{\pi}}(s, a)\pi_s(\mathrm{d}a) \tag{25}$$

$$= \int_{\mathcal{A}} Q_{\hat{\pi}}(s, a)f(A_{\hat{\pi}}(s, a))\hat{\pi}_s(\mathrm{d}a) \tag{26}$$

$$= \int_{\mathcal{A}} Q_{\hat{\pi}}(s, a)f(Q_{\hat{\pi}}(s, a) - V_{\hat{\pi}}(s))\hat{\pi}_s(\mathrm{d}a), \tag{27}$$

where $\pi_s$ and $\hat{\pi}_s$ denote the probability measures corresponding to the distributions $\pi(\cdot \mid s)$ and $\hat{\pi}(\cdot \mid s)$, respectively. Then,

$$\mathbb{E}_{a \sim \pi(\cdot|s)}[Q_{\hat{\pi}}(s, a)] = \int_{\mathcal{A}} Q_{\hat{\pi}}(s, a)f(Q_{\hat{\pi}}(s, a) - V_{\hat{\pi}}(s))\hat{\pi}_s(\mathrm{d}a) \tag{28}$$

$$= \int_{\mathbb{R}} qf(q - V_{\hat{\pi}}(s))\lambda(\mathrm{d}q) \tag{29}$$

$$\geq \left( \int_{\mathbb{R}} q\lambda(\mathrm{d}q) \right) \left( \int_{\mathbb{R}} f(q - V_{\hat{\pi}}(s))\lambda(\mathrm{d}q) \right) \tag{30}$$

$$= \left( \int_{\mathcal{A}} Q_{\hat{\pi}}(s, a)\hat{\pi}_s(\mathrm{d}a) \right) \left( \int_{\mathcal{A}} f(Q_{\hat{\pi}}(s, a) - V_{\hat{\pi}}(s))\hat{\pi}_s(\mathrm{d}a) \right) \tag{31}$$

$$= \left( \int_{\mathcal{A}} Q_{\hat{\pi}}(s, a)\hat{\pi}_s(\mathrm{d}a) \right) \left( \int_{\mathcal{A}} f(A_{\hat{\pi}}(s, a))\hat{\pi}_s(\mathrm{d}a) \right) \tag{32}$$

$$= V_{\hat{\pi}}(s)\mathbb{E}_{a \sim \hat{\pi}(\cdot|s)}[f(A_{\hat{\pi}}(s, a))] \tag{33}$$

$$= V_{\hat{\pi}}(s), \tag{34}$$

where $\lambda$ denotes the pushforward measure of $\hat{\pi}_s$ by $Q_{\hat{\pi}}(s, \cdot)$, and we use Lemma 1 in the third line with $g(x) = 1$ and $h(x) = f(x - V_{\hat{\pi}}(s))$, both of which are non-decreasing. $\qquad\square$

**Lemma 3** (Policy improvement theorem for stochastic policies (Sutton & Barto, 2005; da Silva, 2023))**.** *For any policies $\pi$ and $\hat{\pi}$ satisfying $\mathbb{E}_{a \sim \pi(\cdot|s)}[Q_{\hat{\pi}}(s, a)] \geq V_{\hat{\pi}}(s)$ for all $s \in \mathcal{S}$,*

$$J(\pi) \geq J(\hat{\pi}). \tag{35}$$

*Proof.* This is a straightforward generalization of the policy improvement theorem to stochastic policies. See Section 4.2 of Sutton & Barto (2005) and Theorem 3 of da Silva (2023). □

**Theorem 1** (Policy improvement by reweighting). *Let $\pi, \hat{\pi} : \mathcal{S} \to \Delta(\mathcal{A})$ be policies and $f : \mathbb{R} \to \mathbb{R}$ be a bounded, measurable, non-negative, non-decreasing function. Suppose that $\pi$ satisfies $\pi(a \mid s) \propto f(A_{\hat{\pi}}(s, a))\hat{\pi}(a \mid s)$. Then,*

$$J(\pi) \geq J(\hat{\pi}). \tag{36}$$

*Proof.* Fix $s \in \mathcal{S}$. Let $\pi(a \mid s) = f(A_{\hat{\pi}}(s, a))\hat{\pi}(a \mid s)/Z(s)$, where the normalization function $Z : \mathcal{S} \to \mathbb{R}$ is defined as

$$Z(s) = \int_{\mathcal{A}} f(A_{\hat{\pi}}(s, a))\hat{\pi}_s(\mathrm{d}a). \tag{37}$$

Then, we have

$$1 = \int_{\mathcal{A}} f(A_{\hat{\pi}}(s, a))/Z(s)\hat{\pi}_s(\mathrm{d}a) \tag{38}$$

$$= \mathbb{E}_{a \sim \hat{\pi}(\cdot \mid s)}[f(A_{\hat{\pi}}(s, a))/Z(s)]. \tag{39}$$

Defining $g = f/Z(s)$, we get $\mathbb{E}_{a \sim \hat{\pi}(\cdot \mid s)}[g(A_{\hat{\pi}}(s, a))] = 1$. Since $f$ is non-negative and non-decreasing, so is $g$, and the conclusion directly follows from Lemma 2 (with $\pi(a \mid s) = g(A_{\hat{\pi}}(s, a))\hat{\pi}(a \mid s)$) and Lemma 3. □

**Theorem 2.** *Let $0 \leq w_1 \leq w_2$ be real numbers, $\pi_1, \pi_2, \hat{\pi} : \mathcal{S} \to \Delta(\mathcal{A})$ be policies, and $f : \mathbb{R} \to \mathbb{R}$ be a bounded, measurable, non-negative, non-decreasing function. Suppose that $\pi_i$ satisfies $\pi_i(a \mid s) \propto f(A_{\hat{\pi}}(s, a))^{w_i}\hat{\pi}(a \mid s)$ for $i = 1, 2$. Then,*

$$J(\pi_1) \leq J(\pi_2). \tag{40}$$

*Proof.* Fix $s \in \mathcal{S}$. As in the proof of Theorem 1, write

$$\pi_1(a \mid s) = \frac{f(A_{\hat{\pi}}(s, a))^{w_1}\hat{\pi}(a \mid s)}{Z_1(s)}, \tag{41}$$

$$\pi_2(a \mid s) = \frac{f(A_{\hat{\pi}}(s, a))^{w_2}\hat{\pi}(a \mid s)}{Z_2(s)}, \tag{42}$$

where $Z_1, Z_2 : \mathcal{S} \to \mathbb{R}$ are the normalization functions. Then, we have

$$\pi_2(a \mid s) = f(A_{\hat{\pi}}(s, a))^{w_2 - w_1}\frac{Z_1(s)}{Z_2(s)}\pi_1(a \mid s). \tag{43}$$

Since $Z_1$ and $Z_2$ are both bounded (which follows from the boundedness of $f$), measurable, and non-negative, we can apply Lemma 2 to the bounded, measurable, non-negative, non-decreasing function $x \mapsto f(x)^{w_2 - w_1}Z_1(s)/Z_2(s)$ with $(\pi, \hat{\pi}) = (\pi_2, \pi_1)$ (in the notation of Lemma 2). The result then directly follows from Lemma 3 as before. □

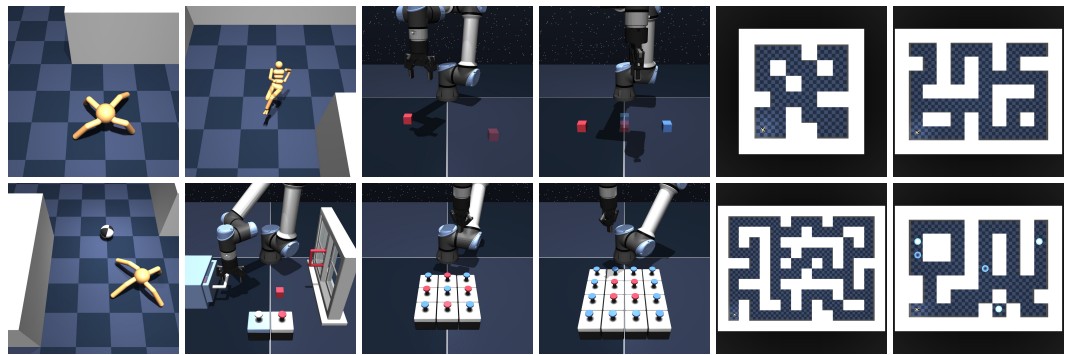

Figure 5: **OGBench environments.**

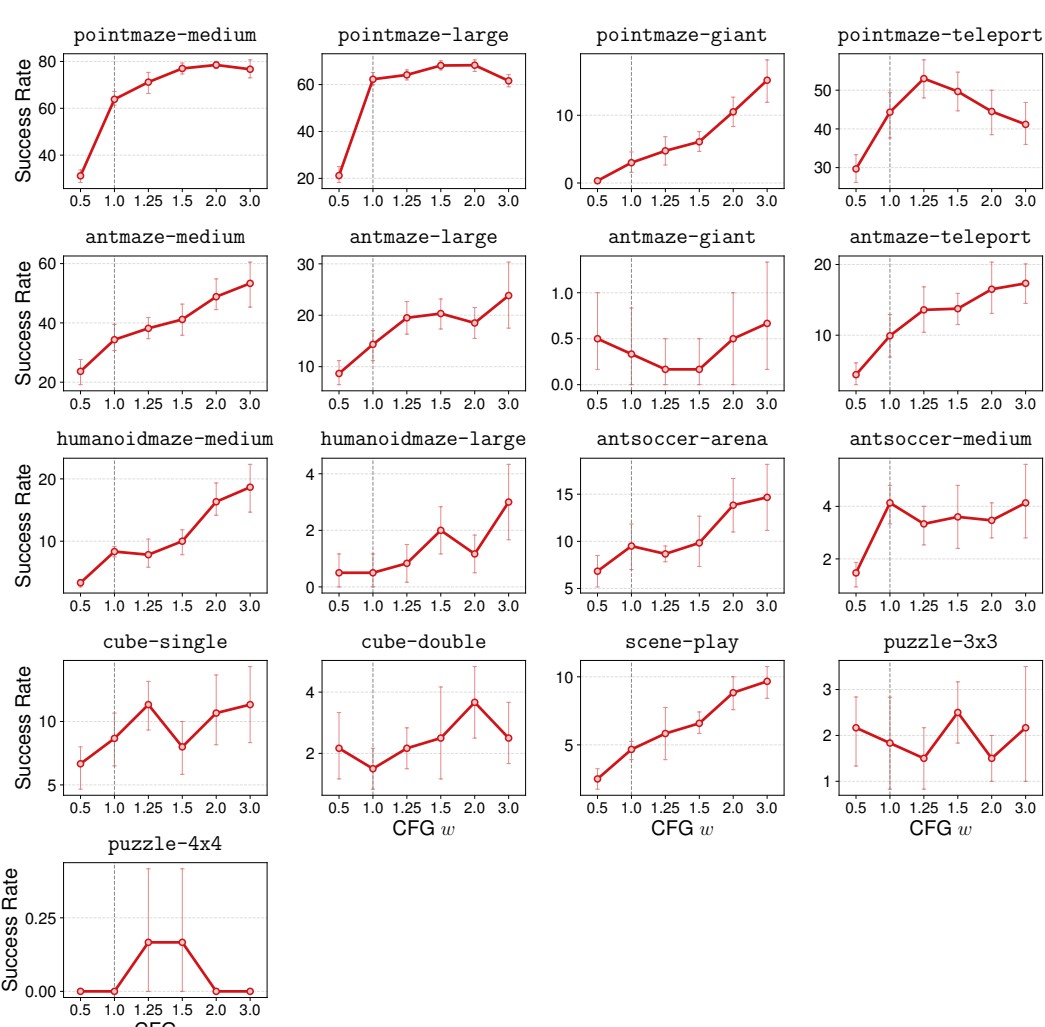

Figure 6: **Full ablation results on CFG weight $w$.** The performance of CFGRL generally improves as the CFG weight increases.

## B ADDITIONAL RESULTS

**Environments.** Figure 5 illustrates OGBench tasks.

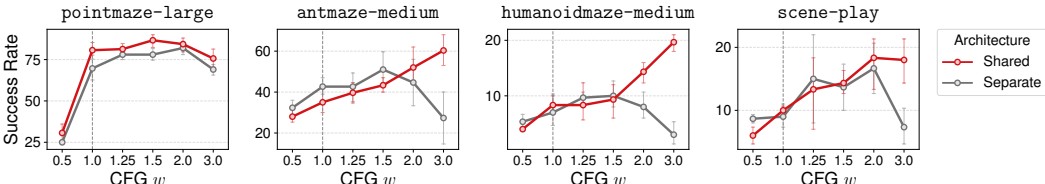

Figure 7: **Ablation study on optimality conditioning.** Shared policies lead to better performance and extrapolation than separate policies, likely because the former shares representations.

**Ablation study on the CFG weight $w$.** We present the full ablation study on the CFG weight $w$ across all 17 state-based OGBench tasks in Figure 6. The results show that the performance improves as the CFG weight increases, although it sometimes declines beyond a certain point, likely because the policy deviates too far from the data distribution.

**Ablation study on optimality conditioning.** When modeling an optimality-conditioned policy $\pi(a \mid s, o)$ with $o \in \{\varnothing, 0, 1\}$, we can either have separate networks for each $o$ value, or share the same network with a learnable optimality embedding. We choose the latter in our experiments, and present an ablation study in Figure 7. The results suggest that the shared architecture generally works and extrapolates better than the separate one. We believe this is likely because extrapolation benefits from shared representations.

## C   IMPLEMENTATION DETAILS

We implement CFGRL on top of the reference implementations provided by OGBench (Park et al., 2025a). Each experiment in this work takes no more than 4 hours on a single A5000 GPU.

**Tasks.** In Section 5, we employ 9 tasks from the ExORL benchmark (Yarats et al., 2022) and 9 single-task (`singletask`) variants from the OGBench suite (Park et al., 2025a). We use the RND datasets for our ExORL experiments. In Section 6, we employ the `oraclerep` variant of OGBench tasks to remove confounding factors related to goal representation learning, where this variant provides ground-truth goal representations (e.g., in `antmaze`, a goal is specified by only the $x$-$y$ position, as opposed to the full 29-dimensional state including proprioceptive information).

**Methods and hyperparameters.** For baselines, we follow the original implementations and hyper-parameters whenever possible (Kostrikov et al., 2022; Park et al., 2025a;b). For GCBC methods in Section 6, we sample goals uniformly from future states, as in the original implementation in OG-Bench (Park et al., 2025a). This can be viewed as an approximation of geometric sampling with a high $\gamma$. We present the full list of the hyperparameters in Tables 4 to 8.

Table 4: **Hyperparameters for ExORL offline RL experiments (Table 1).**

| Hyperparameter | Value |
|---|---|
| Learning rate | 0.0003 |
| Optimizer | Adam (Kingma & Ba, 2015) |
| Gradient steps | 1000000 |
| Minibatch size | 1024 |
| MLP dimensions | $[512, 512, 512]$ |
| Nonlinearity | Mish (Misra, 2020) |
| Target network smoothing coefficient | 0.005 |
| Discount factor $\gamma$ | 0.99 (default), 0.995 (`antmaze-giant`, `humanoidmaze`, `antsoccer`) |
| Flow steps | 32 |
| Flow time sampling distribution | $\mathrm{Unif}([0, 1])$ |
| IQL expectile | 0.9 |
| CFGRL $w$ and AWR $1/\beta$ | Table 5 |

Table 5: **Per-task hyperparameters for ExORL offline RL experiments (Table 4).**

| Task | AWR $1/\beta$ | CFGRL $w$ |
|---|---|---|
| `walker-stand` | 3 | 30 |
| `walker-walk` | 3 | 30 |
| `walker-run` | 10 | 30 |
| `quadruped-walk` | 3 | 3 |
| `quadruped-run` | 3 | 10 |
| `cheetah-run` | 30 | 10 |
| `cheetah-run-backward` | 3 | 30 |
| `jaco-reach-top-right` | 3 | 3 |
| `jaco-reach-top-left` | 3 | 3 |

Table 6: **Hyperparameters for OGBench offline RL experiments (Table 2).**

| Hyperparameter | Value |
|---|---|
| Learning rate | 0.0003 |
| Optimizer | Adam (Kingma & Ba, 2015) |
| Gradient steps | 500000 |
| Minibatch size | 256 |
| MLP dimensions | $[512, 512, 512, 512]$ |
| Nonlinearity | GELU (Hendrycks & Gimpel, 2016) |
| Target network smoothing coefficient | 0.005 |
| Discount factor $\gamma$ | 0.99 (default), 0.995 (`antmaze-giant`, `humanoidmaze`, `antsoccer`) |
| Flow steps | 16 |
| Flow time sampling distribution | $\mathrm{Unif}([0, 1])$ |
| IQL expectile | 0.9 |
| AWR $1/\beta$ and CFGRL $w$ | Table 7 |

Table 7: **Per-task hyperparameters for OGBench offline RL experiments (Table 2).**

| Task | AWR $1/\beta$ | CFGRL $w$ |
|---|---|---|
| `pointmaze-large-navigate` | 10 | 1 |
| `pointmaze-teleport-navigate` | 1 | 1 |
| `antmaze-large-navigate` | 10 | 1.25 |
| `antmaze-teleport-navigate` | 10 | 3 |
| `humanoidmaze-large-navigate` | 3 | 1 |
| `antsoccer-arena-navigate` | 10 | 1.5 |
| `cube-single-play` | 1 | 1.5 |
| `scene-play` | 3 | 3 |
| `puzzle-3x3-play` | 1 | 3 |

Table 8: **Hyperparameters for GCBC experiments (Table 3).**

| Hyperparameter | Value |
|---|---|
| Learning rate | 0.0003 |
| Optimizer | Adam (Kingma & Ba, 2015) |
| Gradient steps | 1000000 |
| Minibatch size | 1024 (states), 256 (pixels) |
| MLP dimensions | $[512, 512, 512, 512]$ |
| Nonlinearity | GELU (Hendrycks & Gimpel, 2016) |
| Image augmentation probability | 0.5 |
| Flow steps | 16 |
| Flow time sampling distribution | $\text{Unif}([0, 1])$ |
| CFGRL $w$ | 3 |
| Subgoal steps for hierarchical BC | 25 (default), 10 (OGBench manipulation), 50 (`humanoidmaze`) |

