# OpenReview forum: "Diffusion Guidance Is a Controllable Policy Improvement Operator"
_ICLR.cc/2026/Conference — Submitted to ICLR 2026_

### Official Review · Reviewer_y8PU · 2025-10-26

**Soundness:** 2
**Presentation:** 3
**Contribution:** 2
**Rating:** 4
**Confidence:** 3

**Summary:**

The paper reframes classifier-free guidance [1] from flow models as a controllable policy-improvement operator in RL, sampling from a prior policy tilted by an optimality term. Empirically, the proposed CFGRL improves over imitation/weighted-extraction baselines such as Goal-conditioned BC and AWR [2].

[1] Ho, Jonathan, and Tim Salimans. "Classifier-free diffusion guidance." arXiv preprint arXiv:2207.12598 (2022).

[2] Peng, Xue Bin, et al. "Advantage-weighted regression: Simple and scalable off-policy reinforcement learning." arXiv preprint arXiv:1910.00177 (2019).

**Strengths:**

1. Theory & clarity.

The theoretical development is solid. The paper cleanly formalizes the link between generative-model guidance and policy improvement, and presents it in a way that is both accessible and precise.

2. Guidance weight $w$: evidence matches the claim.

Figure 4 empirically supports Remark 2: increasing the classifier-free guidance weight $w$ consistently steers the policy toward better directions. The additional results in the Appendix (Fig. 6) further corroborate this trend across settings, strengthening the central guidance-as-control narrative.

3. CFGRL for GCBC: clear setup, convincing gains.

Section 6 is well-specified and persuasive. The experiments substantiate the claim that CFGRL provides a principled, tunable alternative to standard GCBC: whereas prior GCBC effectively fixes $w=1$, the proposed framework exposes $w$ as a controllable knob and yields improved performance accordingly. The alignment between the theoretical framing and the observed gains is compelling.

**Weaknesses:**

1. The paper positions CFGRL as a scalable policy-extraction tool inspired by classifier-free guidance, and shows consistent gains over GCBC/flow-GCBC on OGBench. I think the paper's central thesis is to scale up RL by leveraging the scalability of “modern generative learning,” and, by implication, generative-model-based techniques like CFGRL should match or exceed the performance of RL-based methods such as FQL [3], which are harder to scale under the author's claim (as noted at line 28). To substantiate the broader “scale up RL” thesis, it would be essential to include strong baselines. Including FQL results would clarify where the advantage of CFGRL lies, in competing with modern value-gradient methods rather than with an unexpressive policy-extraction scheme like AWR [4].

[3] Park, Seohong, Qiyang Li, and Sergey Levine. "Flow q-learning." arXiv preprint arXiv:2502.02538 (2025).

[4] Park, Seohong, et al. "Is value learning really the main bottleneck in offline RL?." Advances in Neural Information Processing Systems 37 (2024): 79029-79056.

**Questions:**

1. Positioning vs. SOTA offline RL: strengths and weaknesses.

Even if CFGRL is not intended to replace SOTA offline RL, a clear trade-off analysis is needed. Please add representative value-/policy-gradient baselines (e.g., FQL) under matched protocols, and discuss where CFGRL is advantageous (simplicity, stability, test-time controllability $w$) and where it is weaker (asymptotic performance, OOD robustness, sample efficiency). A brief ablation on wall-clock, hyper-sensitivity, and compute/memory would help practitioners decide when to choose CFGRL as a “designer’s toolbox” component.


2. Task selection in Table 2.

OGBench contains many tasks, but Table 2 reports nine. What criteria determined this subset? Please provide a justification for the selection or expand the evaluation to a broader, representative slice of OGBench to reduce selection bias.


3. Baselines for the sampling/extraction objective (Eq. 8).

Since many policy-extraction schemes target the same tilted distribution, AWR alone is insufficient as a comparator. Please include Relative Trajectory Balance [5] as an additional baseline: it optimizes an equivalent target distribution in theory and achieves stronger performance than AWR on several D4RL tasks [6].

4. Scope beyond $o\in\{0,1,\emptyset\}$: return-conditioned guidance.

Section 4.1 instantiates $o\in\{0,1,\emptyset\}$, but under the Control-as-an-Inference view, the optimality variable naturally extends to learned Q-values or discounted returns. This suggests a return-conditioned variant—analogous to RvS/Decision Transformer/Decision Diffuser [7][8][9]—where classifier-free guidance tilts actions by target return. Please clarify how the $o\in\{0,1,\emptyset\}$ setting in CFGRL compares to return-conditioned BC with classifier-free guidance (both conceptually and empirically).

[5] Venkatraman, Siddarth, et al. "Amortizing intractable inference in diffusion models for vision, language, and control." Advances in neural information processing systems 37 (2024): 76080-76114.

[6] Fu, Justin, et al. "D4RL: Datasets for Deep Data-Driven Reinforcement Learning."

[7] Emmons, Scott, et al. "RvS: What is Essential for Offline RL via Supervised Learning?." International Conference on Learning Representations.

[8] Chen, Lili, et al. "Decision transformer: Reinforcement learning via sequence modeling." Advances in neural information processing systems 34 (2021): 15084-15097.

[9] Ajay, Anurag, et al. "Is Conditional Generative Modeling all you need for Decision Making?." The Eleventh International Conference on Learning Representations.

---

> ### Author Response · Authors · 2025-11-20
>
> Thank you for your detailed review. Please see our response below:
>
> **On comparisons between CFGRL and end-to-end RL methods**.
>
> Intentionally, CFGRL does not require a differentiable Q-function, but rather only samples of Q-values -- this is a benefit of the classifier-*free* nature of the algorithm, and is what allows CFGRL to be used even when a Q-function does not exist (see Section 6). CFGRL outperforms AWR reliably within this constraint.
>
> A alternate set of methods require differentiating through a learned Q-function -- we ran new experiments to compare two examples of such methods, FQL [2] and Q-Score-Matching (QSM) [3]. We use a regularized QSM where each denoising step follows a (tuned) combination of the unconditional flow and the Q-score. In general, we expect that such methods can attain better overall results in proportion to how accurate the learned Q function is. However, such methods also comes with drawbacks -- FQL uses an expensive distillation procedure that requires flow-denoising during training, and QSM requires keeping a Q-function around during evaluation. Additionally, both methods require assume that we have access to a smooth and accurate Q-function.
>
> **We also note that CFGRL does not require tuning a temperature during training, which saves significant compute in practice.** In contrast, AWR and FQL require sweeping over this sensitive parameter. The results shown for AWR and FQL involve training 5 separate models per seed (and selecting the highest performing), whereas CFGRL and QSM only train a single model.
>
> We include an update comparison on representative ExORL tasks:
>
> | | CFGRL | AWR | FQL | QSM |
> |----------------|-------|-----|-----|---------|
> | walker-run | 282 ±6 | 247 ±10 | 516 ± 18 | 469 ± 6 |
> | quadruped-run | 571 ±25 | 485 ±7 | 571 ± 25 | 633 ± 6 |
> | cheetah-run | 216 ±15 | 168 ±7 | 366 ± 26 | 257 ± 10 |
>
> **Section 6 shows that CFGRL is strongly beneficial when we cannot train a Q function, precisely because it does not require gradients through a smooth Q-function.** This limitation is quite often true for many real-world tasks, where we do not have enough data to reliably train a strong Q-function via TD learning. For sake of reference, we have updated the table below to also include GC-IQL, a baseline which does train a goal-conditioned value function. Note that **CFGRL can at times be competitive even versus this full RL method.** This can happen when the prerequisite problem of learning a value function is hard -- for example, in the `visual-antmaze` setting where pixel features must be learned, or `pointmaze` due to its low-dimensional observation space. In these settings, CFGRL may outperform value-learning methods. However, we note that we do not expect non-value-function CFGRL to outperform a full RL pipeline in general.
>
> |                                | CFGRL (No Value) | GC-BC (No Value) | GC-IQL (Using Value Function) |
> |--------------------------------|------------------|------------------|-------------------------------|
> | pointmaze-medium-navigate      | **77 ± 6**       | 9 ± 5            | 53 ± 8                        |
> | pointmaze-large-navigate       | **77 ± 5**       | 25 ± 9           | 34 ± 3                        |
> | pointmaze-giant-navigate       | **30 ± 10**      | 2 ± 2            | 0 ± 0                         |
> | antmaze-medium-navigate        | 53 ± 12          | 25 ± 8           | **71 ± 4**                    |
> | antmaze-large-navigate         | 24 ± 10          | 20 ± 4           | **34 ± 4**                    |
> | antmaze-teleport-navigate      | **35 ± 9**       | 19 ± 5           | **35 ± 5**                    |
> | visual-antmaze-medium-navigate | **23 ± 4**       | 11 ± 5           | 11 ± 1                        |
> | visual-antmaze-large-navigate  | **11 ± 2**       | 5 ± 2            | 4 ± 1                         |
>
> **On task selection**. We have chosen a smaller subset for computational reasons. The subset is not cherry-picked; note that we evaluate on one of the most representative tasks from each “category” on OGBench (pointmaze, stochastic pointmaze (teleport), antmaze, stochastic antmaze (teleport), humanoidmaze, antsoccer, cube, scene, and puzzle).
>
> [1] Peng, Xue Bin, Aviral Kumar, Grace Zhang, and Sergey Levine. "Advantage-weighted regression: Simple and scalable off-policy reinforcement learning." arXiv preprint arXiv:1910.00177 (2019).
>
> [2] Park, Seohong, Qiyang Li, and Sergey Levine. "Flow q-learning." arXiv preprint arXiv:2502.02538 (2025).
>
> [3] Psenka, Michael, Alejandro Escontrela, Pieter Abbeel, and Yi Ma. "Learning a diffusion model policy from rewards via q-score matching." arXiv preprint arXiv:2312.11752 (2023).

---

> > ### Author Response · Authors · 2025-11-20
> >
> > **On return-conditioning vs. binary optimalities**. For classifier-free methods, in the end we need to pick a specific conditioning variable to condition on. Return-conditioned methods tend to choose this number heuristically -- for example in Decision transformer they must specify a desired return, and this is unideal as they search for this manually per environment. Advantage-conditioning is more robust in that an advantage of 0 is generally reasonable across all environments. To show this, we performed a sweep ablating the binary advantage threshold:
> >
> > | $\texttt{Task}$ | $A \geq -0.5$ | $A \geq 0.0$ | $A \geq 0.5$ |
> > |:--------------------------------------:|:-------------------:|:-------------------:|:-------------------:|
> > | $\texttt{pointmaze-large-navigate}$ | $82 {\tiny \pm 10}$ | $98 {\tiny \pm 3}$ | $100 {\tiny \pm 0}$ |
> >  | $\texttt{pointmaze-teleport-navigate}$ | $38 {\tiny \pm 18}$ | $57 {\tiny \pm 12}$ | $50 {\tiny \pm 14}$ |
> > | $\texttt{antmaze-large-navigate}$ | $18 {\tiny \pm 8}$ | $23 {\tiny \pm 21}$ | $7 {\tiny \pm 9}$ |
> > | $\texttt{antmaze-teleport-navigate}$ | $27 {\tiny \pm 5}$ | $10 {\tiny \pm 12}$ | $28 {\tiny \pm 8}$ |
> >  | $\texttt{humanoidmaze-large-navigate}$ | $0 {\tiny \pm 0}$ | $0 {\tiny \pm 0}$ | $2 {\tiny \pm 3}$ |
> > | $\texttt{antsoccer-arena-navigate}$ | $27 {\tiny \pm 17}$ | $27 {\tiny \pm 16}$ | $25 {\tiny \pm 13}$ |
> > | $\texttt{cube-single-play}$ | $83 {\tiny \pm 4}$ | $77 {\tiny \pm 9}$ | $77 {\tiny \pm 9}$ | | $\texttt{scene-play}$ | $17 {\tiny \pm 21}$ | $12 {\tiny \pm 6}$ | $10 {\tiny \pm 9}$ |
> >  | $\texttt{puzzle-3x3-play}$ | $3 {\tiny \pm 4}$ | $3 {\tiny \pm 4}$ | $5 {\tiny \pm 3}$ |
> >
> > The table above (4 seeds with standard deviations) shows that while the exact optimal threshold may vary per-task, a global setting of global setting of $A \geq 0$ is robust enough to capture a majority of the performance.

---

> ### Comment · Reviewer_y8PU · 2025-11-28
>
> I agree that CFGRL is particularly useful in scenarios where training a reliable Q-function is difficult, though this benefit appears limited to goal-conditioned tasks. Regarding computational cost and hyperparameter tuning, while CFGRL avoids the training-time temperature sweeps required by AWR, QSM, or FQL, I am not fully convinced this constitutes a decisive advantage. The community has begun to adopt standard practices for tuning methods like FQL and QSM, and CFGRL itself is not tuning-free. CFGRL still requires selecting the test-time guidance weight $w$, which, although it does not require retraining, must still be searched to obtain optimal performance.
>
> Consequently, in standard settings where Q-learning is feasible and modern consumer-grade compute, such as the RTX 4090, is available, I remain unconvinced that a practitioner should prefer CFGRL over stronger value-gradient baselines like FQL or QSM. The trade-off between the simplicity of CFGRL and the performance gap shown in the new ExORL table suggests that for general offline RL, other Q-based methods remain the superior choice.
>
> The responses regarding return-conditioning vs. binary optimalities were very helpful and resolved my questions.
>
> ---
> While I appreciate the clarifications and the advantage of the method in goal-conditioned settings, my primary concern regarding the positioning and the advantage of CFGRL over state-of-the-art RL methods still remains. Therefore, I will maintain my original score.

---

### Official Review · Reviewer_gz24 · 2025-10-30

**Soundness:** 3
**Presentation:** 3
**Contribution:** 2
**Rating:** 4
**Confidence:** 3

**Summary:**

This paper introduces the CFGRL framework, which establishes a theoretical connection between classifier-Free diffusion guidance and the policy improvement operator in classical reinforcement learning. It formulates the improved policy as a "product policy", defined as a reference policy multiplied by an advantage-based optimality function. The authors highlight that the guidance weight $w$ enables flexible control over the extent of policy improvement at test time, without requiring retraining as in methods such as AWR. The proposed CFGRL framework is validated on both offline RL and goal-conditioned behavior cloning (GCBC) tasks, demonstrating superior performance over AWR and GCBC methods that can be interpreted as using a fixed guidance weight of 1. These findings provide evidence that guidance weights greater than one can effectively cooperate with diffusion models to achieve more efficient policy improvement.

**Strengths:**

1. The paper is well written, and the presentation of results is clear and easy for readers to follow.

2. To the best of my knowledge, this paper is the first to theoretically establish and prove the connection between classifier-free guided diffusion policy sampling and the policy improvement operator in RL.

3. The authors’ analysis of AWR’s weakness in Section 5, together with the experimental observation that CFGRL can sustain larger guidance weights than AWR, constitutes an interesting result.

**Weaknesses:**

1. The main limitation of this paper lies in that most of its ideas have already appeared independently in prior works. For example, the relationship between classifier-free guidance and weighted regression has been discussed in [1], while the use of classifier-free guidance for policy improvement and the adjustment of different guidance weights was explored in [2]. Although the authors argue that [2] focuses on generating future state sequences whereas CFGRL generates single-step actions, I consider this distinction in output space relatively trivial. Apart from these aspects, the most notable contribution is the explicit formulation of the connection between the **policy improvement operator** and **classifier-free guided diffusion policy sampling**. However, this result is not particularly surprising, and the theoretical derivation itself is fairly straightforward. Considering this as the paper's primary contribution, it remains questionable whether the work alone is sufficient to support publication at ICLR.

2. Moreover, the baseline methods in the experiments are relatively weak, and the proposed algorithm does not obtain a significant performance advantage under these settings, which limits the overall algorithmic contribution. Although the authors emphasize that "By itself, CFGRL does not represent a state-of-the-art RL algorithm, but rather an additional tool in the algorithm designer's toolbox," the paper would be more compelling if the authors could further demonstrate how innovative algorithms can be derived using the CFGRL framework.

[1] Ho, Jonathan, and Tim Salimans. "Classifier-free diffusion guidance." *arXiv preprint arXiv:2207.12598* (2022).

[2] Ajay, Anurag, et al. "Is Conditional Generative Modeling all you need for Decision Making?." *The Eleventh International Conference on Learning Representations*.

**Questions:**

1. What kinds of more advanced algorithms do the authors expect could be derived from the CFGRL framework, beyond current implementations that largely reproduce methods already proposed in prior work?

2. Beyond GCBC and offline RL, what other problem domains could the CFGRL framework be applied to?

---

> ### Author Response · Authors · 2025-11-20
>
> Thank you for your detailed review. Please see our response below:
>
> **On CFGRL vs. decision-diffuser**. We argue that the distinction between modelling trajectories and actions is nontrivial. Decision diffuser uses a network with 12 dense layers, an effective dimensionality of 256*20, a separate inverse kinematics model MLP, and 100 denoising steps. We only use a single 512-dim MLP with 4 layers and 16 steps, which is computationally cheaper by an order of > 100x. Decision-diffuser also does not draw a connection to policy improvement. Thus, we believe these are widely different settings worthy of separate study.
>
> **On comparisons to end-to-end RL methods**.
> Section 6 shows that CFGRL is strongly beneficial when we cannot train a Q function. This limitation is quite often true for many real-world tasks, where we do not have enough data to reliably train a strong Q-function via TD learning.
>
> For sake of reference, we have updated the table below to also include GC-IQL, a baseline which does train a goal-conditioned value function. Note that **CFGRL can at times be competitive even versus this full RL method.** This can happen when the prerequisite problem of learning a value function is hard -- for example, in the `visual-antmaze` setting where pixel features must be learned, or `pointmaze` due to its low-dimensional observation space. In these settings, CFGRL may outperform value-learning methods. However, we note that we do not expect non-value-function CFGRL to outperform a full RL pipeline in general.
>
> |                                | CFGRL (No Value) | GC-BC (No Value) | GC-IQL (Using Value Function) |
> |--------------------------------|------------------|------------------|-------------------------------|
> | pointmaze-medium-navigate      | **77 ± 6**       | 9 ± 5            | 53 ± 8                        |
> | pointmaze-large-navigate       | **77 ± 5**       | 25 ± 9           | 34 ± 3                        |
> | pointmaze-giant-navigate       | **30 ± 10**      | 2 ± 2            | 0 ± 0                         |
> | antmaze-medium-navigate        | 53 ± 12          | 25 ± 8           | **71 ± 4**                    |
> | antmaze-large-navigate         | 24 ± 10          | 20 ± 4           | **34 ± 4**                    |
> | antmaze-teleport-navigate      | **35 ± 9**       | 19 ± 5           | **35 ± 5**                    |
> | visual-antmaze-medium-navigate | **23 ± 4**       | 11 ± 5           | 11 ± 1                        |
> | visual-antmaze-large-navigate  | **11 ± 2**       | 5 ± 2            | 4 ± 1                         |
>
> We believe these additional comparisons make the strengths of CFGRL more clear. Please let us know if there are still questions.
>
> With regards to your questions on "what other problem domains could the CFGRL framework be applied to", we emphasize that the main strength of CFGRL is its simplicity and theoretical soundness. There is precedent in RL literature of such components being useful without state-of-the-art results. For example, the AWR method [1] has been used and extended very widely, although AWR itself did not outperform the strongest baselines, and its main virtue (like with CFGRL) was its simplicity and the ease with which it could be applied to a wide variety of different policy classes.
>
> One motivating example is in practical settings such as robotic control. Due to the complexity of the environment, it is standard practice to use goal-conditioned methods [2] that avoid explicit Q-learning. CFGRL can likely be directly applied in these settings, as evidenced by the above table, CFGRL can typically be a drop-in replacement to outperform GCBC, and does not require a learned Q function. A rich direction of future work can examine the precise optimality conditioning scheme, or further extend goal-conditioned optimalities to a richer subset (i.e. masked subgoals, successor features, etc) within the CFGRL framework.
>
> [1] Peng, Xue Bin, Aviral Kumar, Grace Zhang, and Sergey Levine. "Advantage-weighted regression: Simple and scalable off-policy reinforcement learning." arXiv preprint arXiv:1910.00177 (2019).
>
> [2] Black, Kevin, Mitsuhiko Nakamoto, Pranav Atreya, Homer Walke, Chelsea Finn, Aviral Kumar, and Sergey Levine. "Zero-shot robotic manipulation with pretrained image-editing diffusion models." arXiv preprint arXiv:2310.10639 (2023).

---

### Official Review · Reviewer_6iHH · 2025-10-31

**Soundness:** 3
**Presentation:** 4
**Contribution:** 3
**Rating:** 4
**Confidence:** 3

**Summary:**

This paper proposes Classifier-Free Guided Reinforcement Learning (CFGRL), which interprets the classifier-free guidance (CFG) mechanism of diffusion models as a policy improvement operator.
The authors show that by treating the optimality condition as a discrete binary variable $o\in \{0,1\}$, CFG can be reinterpreted as applying an advantage-weighted transformation on a reference policy. This allows controllable policy improvement via the guidance weight $w$, similar in spirit to temperature or KL coefficients in regularized RL. Experiments demonstrate that CFGRL improves over Advantage-Weighted Regression (AWR) and Goal-Conditioned Behavioral Cloning (GCBC) across offline RL benchmarks.
- An LLM was used to improve writing.

**Strengths:**

1. Simplicity and practical appeal – The method requires only standard diffusion training and allows tuning the improvement strength $w$ at inference, offering a practical way to control policy quality without retraining.

2. Solid empirical demonstration – Results on offline RL and goal-conditioned control tasks consistently show improvements over strong baselines such as AWR and GCBC.

3. Readable and well-presented – The paper is clearly written, with theoretical and empirical sections well balanced.

**Weaknesses:**

1. Limited novelty beyond reinterpretation

The core idea—recasting classifier-free guidance as a policy improvement operator—is conceptually elegant but incremental.

The method mainly replaces the continuous classifier (score function) in diffusion guidance with a discrete optimality variable, which is a small modification rather than a fundamentally new algorithmic contribution.

Much of the theoretical framing follows directly from existing formulations of advantage-weighted regression and control-as-inference.

2. Lack of comparative analysis with continuous guidance

The paper would be significantly stronger if it empirically compared continuous value-based guidance (e.g., by Q-values or advantages) versus discrete optimality-based guidance.

Such a comparison could clarify what is actually gained by discretizing optimality in this setting.

Without this, CFGRL appears as a special case of prior decision-diffuser-style methods using advantage-conditioned diffusion.

**Questions:**

See weaknesses.

---

> ### Author Response · Authors · 2025-11-20
>
> Thank you for your detailed review. Please see our response below:
>
> First, we wish to re-iterate that we do not present CFGRL as a full end-to-end method, but rather a surprisingly strong component that interprets policy extraction through modern generative models (i.e. diffusion models) in simple, theoretically solid way. There is precedent in RL literature of such components being useful without state-of-the-art results. For example, the AWR method [1] has been used and extended very widely, although AWR itself did not outperform the strongest baselines, and its main virtue (like with CFGRL) was its simplicity and the ease with which it could be applied to a wide variety of different policy classes.
>
> **On CFGRL vs. decision-diffuser**. We note that the distinction between modelling trajectories and actions is nontrivial. Decision diffuser uses a network with 12 dense layers, an effective dimensionality of 256*20, a separate inverse kinematics model MLP, and 100 denoising steps. We only use a single 512-dim MLP with 4 layers and 16 steps, which is computationally cheaper by an order of > 100x.
>
> **On discrete vs. continuous guidance**. We utilize discrete optimality in CFGRL because it is necessary to derive a classifier-*free* guidance, as described in Equation 10. This enables **CFGRL to only requires samples from a value function**, not smooth gradients. This is the same requirement as AWR, and we show that CFGRL outperforms AWR reliably within this constraint.
>
> A alternate set of methods require *differentiating through a learned Q-function* -- we ran new experiments to compare two examples of such methods, FQL [2] and Q-Score-Matching (QSM) [3]. We use a regularized QSM where each denoising step follows a (tuned) combination of the unconditional flow and the Q-score. In general, we expect that such methods can attain better overall results in proportion to how accurate the learned Q function is. However, such methods also comes with drawbacks -- FQL uses an expensive distillation procedure that requires flow-denoising during training, and QSM requires keeping a Q-function around during evaluation. Additionally, both methods require assume that we have access to a smooth Q-function.
>
> **We also note that CFGRL does not require tuning a temperature during training, which saves significant compute in practice.** In contrast, AWR and FQL require sweeping over this sensitive parameter. The results shown for AWR and FQL involve training 5 separate models per seed (and selecting the highest performing), whereas CFGRL and QSM only train a single model.
>
> We include an update comparison on representative ExORL tasks:
>
> | | CFGRL (Q samples) | AWR (Q samples) | FQL (Q gradient) | QSM  (Q gradient)|
> |----------------|-------|-----|-----|---------|
> | walker-run | 282 ±6 | 247 ±10 | 516 ± 18 | 469 ± 6 |
> | quadruped-run | 571 ±25 | 485 ±7 | 571 ± 25 | 633 ± 6 |
> | cheetah-run | 216 ±15 | 168 ±7 | 366 ± 26 | 257 ± 10 |
>
> **Section 6 shows that CFGRL is strongly beneficial when we cannot train a Q function, precisely because discrete optimality does not require gradients through a smooth Q-function.** This limitation is quite often true for many real-world tasks, where we do not have enough data to reliably train a strong Q-function via TD learning. For sake of reference, we have updated the table below to also include GC-IQL, a baseline which does train a goal-conditioned value function. Note that **CFGRL can at times be competitive even versus this full RL method.** This can happen when the prerequisite problem of learning a value function is hard -- for example, in the `visual-antmaze` setting where pixel features must be learned, or `pointmaze` due to its low-dimensional observation space. In these settings, CFGRL may outperform value-learning methods. However, we note that we do not expect non-value-function CFGRL to outperform a full RL pipeline in general.
>
> |                                | CFGRL (No Value) | GC-BC (No Value) | GC-IQL (Using Value Function) |
> |--------------------------------|------------------|------------------|-------------------------------|
> | pointmaze-medium-navigate      | **77 ± 6**       | 9 ± 5            | 53 ± 8                        |
> | pointmaze-large-navigate       | **77 ± 5**       | 25 ± 9           | 34 ± 3                        |
> | pointmaze-giant-navigate       | **30 ± 10**      | 2 ± 2            | 0 ± 0                         |
> | antmaze-medium-navigate        | 53 ± 12          | 25 ± 8           | **71 ± 4**                    |
> | antmaze-large-navigate         | 24 ± 10          | 20 ± 4           | **34 ± 4**                    |
> | antmaze-teleport-navigate      | **35 ± 9**       | 19 ± 5           | **35 ± 5**                    |
> | visual-antmaze-medium-navigate | **23 ± 4**       | 11 ± 5           | 11 ± 1                        |
> | visual-antmaze-large-navigate  | **11 ± 2**       | 5 ± 2            | 4 ± 1                         |

---

> > ### Author Response · Authors · 2025-11-20
> >
> > [1] Peng, Xue Bin, Aviral Kumar, Grace Zhang, and Sergey Levine. "Advantage-weighted regression: Simple and scalable off-policy reinforcement learning." arXiv preprint arXiv:1910.00177 (2019).
> >
> > [2] Park, Seohong, Qiyang Li, and Sergey Levine. "Flow q-learning." arXiv preprint arXiv:2502.02538 (2025).
> >
> > [3] Psenka, Michael, Alejandro Escontrela, Pieter Abbeel, and Yi Ma. "Learning a diffusion model policy from rewards via q-score matching." arXiv preprint arXiv:2312.11752 (2023).

---

### Official Review · Reviewer_Bfm1 · 2025-11-01

**Soundness:** 3
**Presentation:** 3
**Contribution:** 3
**Rating:** 6
**Confidence:** 2

**Summary:**

This paper proposes CFGRL, a simple, controllable policy-improvement operator that leverages classifier-free guidance from diffusion/flow models. Policies are factorized as a product of a reference policy and an “optimality” term that is a monotone function of the advantage; sampling from this product is achieved by composing unconditional and optimality-conditioned policy scores, with a test-time guidance weight $w$ controlling the strength of improvement. The authors prove that such product policies improve over the reference and that increasing $w$ yields further improvement (with the usual trade-off against distribution shift). They also show that, under certain choices, guided sampling matches the solution to a KL-regularized policy-improvement objective. Practically, they instantiate CFGRL with a single diffusion/flow network and provide simple training/sampling algorithms

**Strengths:**

The central insight—viewing policy improvement as classifier-free guidance over an advantage-conditioned policy—is elegant. It unifies guided diffusion sampling with KL-regularized policy improvement and control-as-inference via a clean product-policy view, and shows that test-time guidance directly tunes the improvement strength.

The theory is tidy. The paper also avoids learning an explicit optimality predictor via a Bayes inversion that merges unconditional and optimality-conditioned policies into a single network. Algorithms are minimal and clear.

Empirically, CFGRL improves over AWR on a strong majority of ExORL and OGBench tasks and shows a more favorable scaling trend than AWR’s temperature sweep. The GCBC “upgrade” is impactful: simply guiding the goal-conditioned policy (no value function) yields sizable gains, including hierarchical variants. Code is released; runs are modestly resource-bound.

**Weaknesses:**

The paper notes that larger w both improves $ A_{ \hat \pi }$ and deviates more from the dataset policy, possibly hurting performance; the ablation indeed shows performance sometimes declines past a point, but there’s no adaptive or trust-region control of $w$ or measured KL to the prior.

For the offline RL part, results are averaged over four seeds; gains are consistent but sometimes modest. The GCBC part uses more seeds, but a wider set of domains and stronger end-to-end RL baselines would further cement significance. The paper itself notes it is not a full SOTA RL algorithm.

Flow steps (16–32) suggest non-trivial inference cost relative to a single-shot policy; wall-clock sampling latency and throughput are not measured, which matters for deployment.

**Questions:**

Could you add comparisons to CRR/CCR, IDQL with diffusion policy extraction, and Q-score-matching / rejection-sampling approaches, ideally normalizing compute and tuning budgets? This would position CFGRL more clearly among diffusion-policy extractors.

---

> ### Author Response · Authors · 2025-11-20
>
> Thank you for your detailed review. Please see our response below:
>
> First, we wish to re-iterate that we do not present CFGRL as a full end-to-end method, but rather a surprisingly strong component that interprets policy extraction through modern generative models (i.e. diffusion models) in simple, theoretically solid way. There is precedent in RL literature of such components being useful without state-of-the-art results. For example, the AWR method [1] has been used and extended very widely, although AWR itself did not outperform the strongest baselines, and its main virtue (like with CFGRL) was its simplicity and the ease with which it could be applied to a wide variety of different policy classes.
>
> **Section 5 shows that CFGRL is a reasonable policy extraction method when given a value function.** Notably, CFGRL only requires samples from a value function, the same requirement as AWR, and we show that CFGRL outperforms AWR reliably within this constraint.
>
> A alternate set of methods require differentiating through a learned Q-function -- we ran new experiments to compare two examples of such methods, FQL [2] and Q-Score-Matching (QSM) [3].We use a regularized QSM where each denoising step follows a (tuned) combination of the unconditional flow and the Q-score. In general, we expect that such methods can attain better overall results in proportion to how accurate the learned Q function is. However, such methods also comes with drawbacks -- FQL uses an expensive distillation procedure that requires flow-denoising during training, and QSM requires keeping a Q-function around during evaluation. Additionally, both methods require assume that we have access to a smooth Q-function.
>
> **We also note that CFGRL does not require tuning a temperature during training, which saves significant compute in practice.** In contrast, AWR and FQL require sweeping over this sensitive parameter. The results shown for AWR and FQL involve training 5 separate models per seed (and selecting the highest performing), whereas CFGRL and QSM only train a single model.
>
> We include an update comparison on representative ExORL tasks:
>
> | | CFGRL | AWR | FQL | QSM |
> |----------------|-------|-----|-----|---------|
> | walker-run | 282 ±6 | 247 ±10 | 516 ± 18 | 469 ± 6 |
> | quadruped-run | 571 ±25 | 485 ±7 | 571 ± 25 | 633 ± 6 |
> | cheetah-run | 216 ±15 | 168 ±7 | 366 ± 26 | 257 ± 10 |
>
> **Section 6 shows that CFGRL is strongly beneficial when we cannot train a Q function, precisely because it does not require gradients through a smooth Q-function.** This limitation is quite often true for many real-world tasks, where we do not have enough data to reliably train a strong Q-function via TD learning. For sake of reference, we have updated the table below to also include GC-IQL, a baseline which does train a goal-conditioned value function. Note that **CFGRL can at times be competitive even versus this full RL method.** This can happen when the prerequisite problem of learning a value function is hard -- for example, in the `visual-antmaze` setting where pixel features must be learned, or `pointmaze` due to its low-dimensional observation space. In these settings, CFGRL may outperform value-learning methods. However, we note that we do not expect non-value-function CFGRL to outperform a full RL pipeline in general.
>
> We hope this addresses your request for "stronger end-to-end RL baselines [in] the GCBC part".
>
> |                                | CFGRL (No Value) | GC-BC (No Value) | GC-IQL (Using Value Function) |
> |--------------------------------|------------------|------------------|-------------------------------|
> | pointmaze-medium-navigate      | **77 ± 6**       | 9 ± 5            | 53 ± 8                        |
> | pointmaze-large-navigate       | **77 ± 5**       | 25 ± 9           | 34 ± 3                        |
> | pointmaze-giant-navigate       | **30 ± 10**      | 2 ± 2            | 0 ± 0                         |
> | antmaze-medium-navigate        | 53 ± 12          | 25 ± 8           | **71 ± 4**                    |
> | antmaze-large-navigate         | 24 ± 10          | 20 ± 4           | **34 ± 4**                    |
> | antmaze-teleport-navigate      | **35 ± 9**       | 19 ± 5           | **35 ± 5**                    |
> | visual-antmaze-medium-navigate | **23 ± 4**       | 11 ± 5           | 11 ± 1                        |
> | visual-antmaze-large-navigate  | **11 ± 2**       | 5 ± 2            | 4 ± 1                         |
>
>
> [1] Peng, Xue Bin, Aviral Kumar, Grace Zhang, and Sergey Levine. "Advantage-weighted regression: Simple and scalable off-policy reinforcement learning." arXiv preprint arXiv:1910.00177 (2019).
>
> [2] Park, Seohong, Qiyang Li, and Sergey Levine. "Flow q-learning." arXiv preprint arXiv:2502.02538 (2025).
>
> [3] Psenka, Michael, Alejandro Escontrela, Pieter Abbeel, and Yi Ma. "Learning a diffusion model policy from rewards via q-score matching." arXiv preprint arXiv:2312.11752 (2023).

---

### Meta-Review · Area_Chair_uiAA · 2025-12-06

**Summary:**

This paper proposes CFGRL, a simple, controllable policy-improvement operator that leverages classifier-free guidance from diffusion/flow models. Policies are factorized as a product of a reference policy and an “optimality” term that is a monotone function of the advantage; sampling from this product is achieved by composing unconditional and optimality-conditioned policy scores, with a test-time guidance weight controlling the strength of improvement. The authors prove that such product policies improve over the reference policy. They also show that, under certain choices, guided sampling matches the solution to a KL-regularized policy-improvement objective. Practically, they instantiate CFGRL with a single diffusion/flow network and provide simple training/sampling algorithms.

The paper received generally negative reviews. The core concerns from reviewers concentrated on the novelty issue and the lack of comparison with stronger baselines. But the proposed method does have the merit of being scalable when tuning large diffusion policies, especially for the embodied AI scenarios. Due to the outstanding concerns from reviewers, I'm currently suggesting rejection, but will not be upset if the paper gets accepted.

**Reviewer Concerns:**

Some concerns that I think are still not well-addressed:
- Novelty issue (Reviewer 6iHH, gz24)
- Statistical significance issue with only 4 seeds (Reviewer Bfm1)
- Need extra strong baselines for comparison (Reviewer Bfm1, 6iHH, gz24,y8PU)

**Reviewer Scores:**

I think Reviewer 6iHH might consider increasing his/her score. Reviewer gz24 and y8PU are likely to maintain their scores, as their core concerns (novelty issue and comparison with stronger baselines are not well-addressed).

---

### Decision · Program_Chairs · 2026-01-26

Reject